# Global regulation via modulation of ribosome pausing by the ABC-F protein EttA

Farès Ousalem[1,3], Saravuth Ngo[1], Thomas Oïffer [1], Amin Omairi-Nasser[1], Marion Hamon[2], Laura Monlezun [1] & Grégory Boël [1]✉

Having multiple rounds of translation of the same mRNA creates dynamic complexities along with opportunities for regulation related to ribosome pausing and stalling at specific sequences. Yet, mechanisms controlling these critical processes and the principles guiding their evolution remain poorly understood. Through genetic, genomic, physiological, and biochemical approaches, we demonstrate that regulating ribosome pausing at specific amino acid sequences can produce ~2-fold changes in protein expression levels which strongly influence cell growth and therefore evolutionary fitness. We demonstrate, both in vivo and in vitro, that the ABC-F protein EttA directly controls the translation of mRNAs coding for a subset of enzymes in the tricarboxylic acid (TCA) cycle and its glyoxylate shunt, which modulates growth in some chemical environments. EttA also modulates expression of specific proteins involved in metabolically related physiological and stress-response pathways. These regulatory activities are mediated by EttA rescuing ribosomes paused at specific patterns of negatively charged residues within the first 30 amino acids of nascent proteins. We thus establish a unique global regulatory paradigm based on sequence-specific modulation of translational pausing.

The ATP binding protein F (ABC-F) family, belonging to the ABC superfamily[1], constitutes a group of proteins where all the members studied to date are involved in ribosome-related functions[2–16]. They are widespread in bacteria and eukaryotes and in general, paralogues are present in the same organism, such as *Escherichia coli* which has four ABC-F proteins: EttA, Uup, YheS, and YbiT[2,8,16–18].

ABC-F proteins are composed of two ABC domains in tandem connected together by a linker region that contains the P-site tRNA Interaction Motif (PtIM)[2,9,16,18,19], a signature motif of the family defined by the Pfam database[20] as ABC_tran_Xtn (PF12848). Each ABC-domain carries consensus Walker A and B motifs and in the presence of ATP the two domains come together, binding two ATP molecules to shape an ATP-bound closed conformation[16,18,21]. In this conformation, ABC-F proteins bind to the E site of the 70S ribosome[2,3,5,11,12,14,15] and require the hydrolysis of the ATP molecules to dissociate from the ribosome[2]. Consequently, an ATPase-deficient mutant, in which the two catalytic

glutamates of each Walker B motif of the ABC domain are replaced by glutamines (EQ$_2$ mutant)[21], forms a stable complex with the ribosome[2,3,5,8,14,15]. This stabilization leads to an inhibition of protein synthesis and an arrest of bacterial growth[2,8,17]. To date, twelve cryo-electron microscopy structures of ABC-F proteins in complex with the 70S ribosome are known. These structures reveal an overall geometry with similar contacts between the protein and the ribosome, but the corresponding regions of each ABC-F that contact the ribosome differ in sequences[3,5,7,11,14,15,18,22]. The PtIM extension points towards the peptidyl transferase center (PTC) of the ribosome but shows considerable variation in length[5,11,12,14–16,23]. With the EF-P/eIF-5A protein that has been identified to rescue ribosomes stalled during the synthesis of polyproline-containing proteins[24–26], ABC-F proteins are the only factors known to bind to the ribosomal E site.

Some ABC-F have a clear physiological function such as the antibiotic resistance ABC-F (ARE ABC-F) factors[4–7,9–15,27] which provide

[1]Expression Génétique Microbienne, CNRS, Université Paris Cité, Institut de Biologie Physico-Chimique, Paris, France. [2]CNRS, Institut de Biologie Physico-Chimique, Plateforme de Protéomique, FR550, Paris, France. [3]Present address: Biomarqueurs et nouvelles cibles thérapeutiques en oncologie, INSERM U981, Université Paris Saclay, Institut de Cancérologie Gustave Roussy, Villejuif Cedex, France. ✉e-mail: boel@ibpc.fr

resistance toward antibiotics that target the PTC and the nascent peptide exit tunnel (NPET) of the ribosome[4,5,7,11,12,14,15]. However, for most of the family, their physiological function remains unclear. In eukaryotes, ABC-F proteins are associated with pleiotropic effects. In yeast, the N-terminal domain of GCN20 is involved in the regulation of translation upon amino-acid starvation[28,29]. The second yeast ABC-F, ARB1, is an essential gene important for ribosome biogenesis[30] and the corresponding protein has been identified in the cryo-EM structure of the 60S ribosomal quality control complex in a pre-peptidyl-tRNA cleavage state[23]. The human ABC50 (ABCF1) protein influences translation initiation in vitro at an internal ribosome entry site of mRNA[31] and also functions as an E2 Ubiquitin-conjugating enzyme in the regulation of the inflammatory response in macrophages[32]. The three human ABC-F paralogues have been detected in processes involved in immune responses[33–38] and cancer developments and treatments[36,39–46], but their mechanism of action is unknown. In prokaryotes, all the structurally studied ABC-F can adjust the conformation of the PTC and/or the positioning of the P-site tRNA[8,18] (and some affect bacterial fitness[17]). Based on biochemical and structural studies of the EttA protein, we previously proposed a model in which EttA regulates the first step of translation depending on the ADP:ATP ratio in the reaction[2,3] and showed that loss of EttA resulted in reduced competitive fitness in stationary phase[2]. However, important questions remained: what is the physiological role of EttA in *E. coli*, and does its action impact the translation of all mRNA?

In this study, we have used an unbiased approach to determine the impact of the *ettA* deletion (Δ*ettA*) on the physiology of *E. coli*. The gene deletion creates a hypersensitivity to salt when the bacteria are grown on carbon sources metabolized by the TCA cycle. This phenotype is due to the reduced translation of several genes of the TCA cycle and of genes involved in stress responses. For eight of them, we show that the reduced synthesis occurs during the translation of their mRNAs when the newly polymerized protein contains acidic amino acids within the first 30 amino acids *i.e.* before the elongating peptide reaches the end of the ribosomal NPET. We demonstrate that changes in the expression of *aceB* and *aceA* can explain most of the phenotypes we observed. In addition, we show that the overexpression of *mgtA* gene (encoding a Mg²⁺ transporter) in the Δ*ettA* strain is due to the action of EttA on the Intrinsic Ribosome Destabilization (IRD) effect of the leader peptide MgtL which regulates *mgtA* expression[3].

## Loss of *ettA* impairs bacterial metabolic adaptation and alters protein expression patterns

Our investigation of the physiological function of EttA started with the previously reported phenotype of competitive fitness defect of the *ettA* deleted strain (Δ*ettA*) during the extended stationary phase in LB[2]. In our replication of the experiment, we discerned the phenotype's dependency on the source of the LB medium used (Supplementary Fig. 1a). Subsequently, we have now developed a MOPS-Tricine buffered minimal medium where the phenotype is enhanced, first by reducing the amount of inorganic phosphate, then by using amino acids (aa) as carbon source (MMAA medium) and finally by adding NaCl (0.4 M) to the medium (MMAA-NaCl). In this last condition, there was a rapid loss in competitive fitness of the Δ*ettA* strain compared to the previous condition (1 day vs 3 days) (Supplementary Fig. 1a). In this work we have systematically compared three strains MG1655 WT, Δ*ettA,* and the Δ*ettA* complemented with an exogenous chromosomal copy of *ettA* (C*ettA*). In the MMAA medium, all three strains doubled every 3 h, whereas in the MMAA-NaCl medium, the doubling time of the WT and the complemented strains increased to 8 h (Fig. 1a) while that of the Δ*ettA* strain was more than 9 h and exhibited a much longer lag time (5 h) compared to the two other strains (Fig. 1a). Strains deleted for the other *E. coli* ABC-F genes, grew like the WT, showing that this phenotype is specific to *ettA* (Supplementary Fig. 1b). Noticeably, the addition of NaCl induces EttA expression as

demonstrated using a translational *yfp* (Yellow Fluorescent Protein) reporter fused to *ettA* gene (Fig. 1b and Supplementary Fig. 1c). The *ettA* gene promotor is annotated as an RpoS promoter[47,48], therefore its induced expression in presence of salt may be due to the RpoS sigma factor.

Since the phenotype of the Δ*ettA* strain arises when aa, which are primarily metabolized by the TCA cycle[49], are used as carbon source,

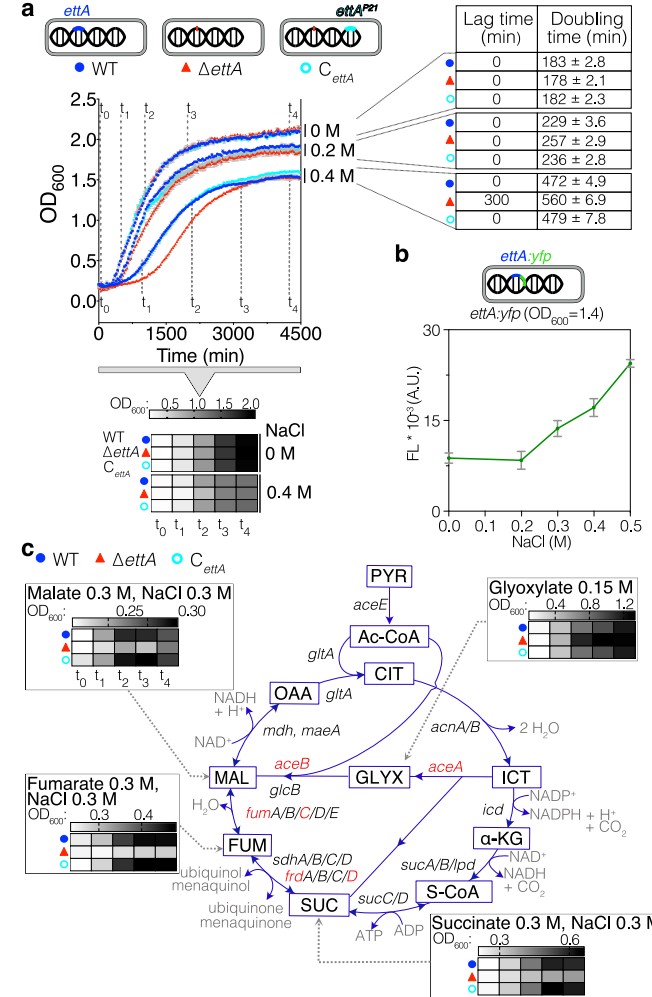

**Fig. 1 | Loss of *ettA* impacts the TCA and glyoxylate shunt pathways. a** Growth curves of the WT, Δ*ettA*, and C*ettA* strains in MMAA medium without or with NaCl (0.2 and 0.4 M). Error bars represent mean ± standard deviation (s.d.) for triplicate experiments. Above and below the curves, heatmaps showing the absorbance (OD₆₀₀) of cultures at various phases of the growth of the WT strain ($t_0$ to $t_4$). The dotted lines indicate the OD values assigned to the heatmap. Right: Table representing the difference in lag phase of Δ*ettA* versus WT and C*ettA* strains and their respective doubling times in the same media calculated as described in the "Methods" section. **b** Graph showing the fluorescence intensity at OD₆₀₀ = 1.4 of the MG1655 *ettA:yfp* strain expressing *ettA* in a translational fusion with a *yfp* gene, in the MMAA medium at different NaCl concentrations (0, 0.2, 0.3, 0.4 or 0.5 M). Background fluorescence of a culture without YFP has been subtracted, the error bars represent mean ± s.d. for triplicate experiments. **c** Representation of the TCA cycle and the glyoxylate shunt pathways showing intermediary products (in boxes), genes coding the enzymes (next to the reaction arrows), and cofactors (in gray). Genes in red were identified in the proteomic study as less expressed in the Δ*ettA* strain (Fig. 2b). Gray heatmaps show the absorbance (OD₆₀₀) of cultures of WT, Δ*ettA*, and C*ettA* strains, at the same growth stage as (**a**), for cultures on different carbon sources at a concentration of 0.3 M (malate, fumarate, and succinate) or 0.15 M (glyoxylate). Scales are presented on the top of each heatmap. Full growth curves and time points used for the heatmap are presented in Supplementary Fig. 1e.

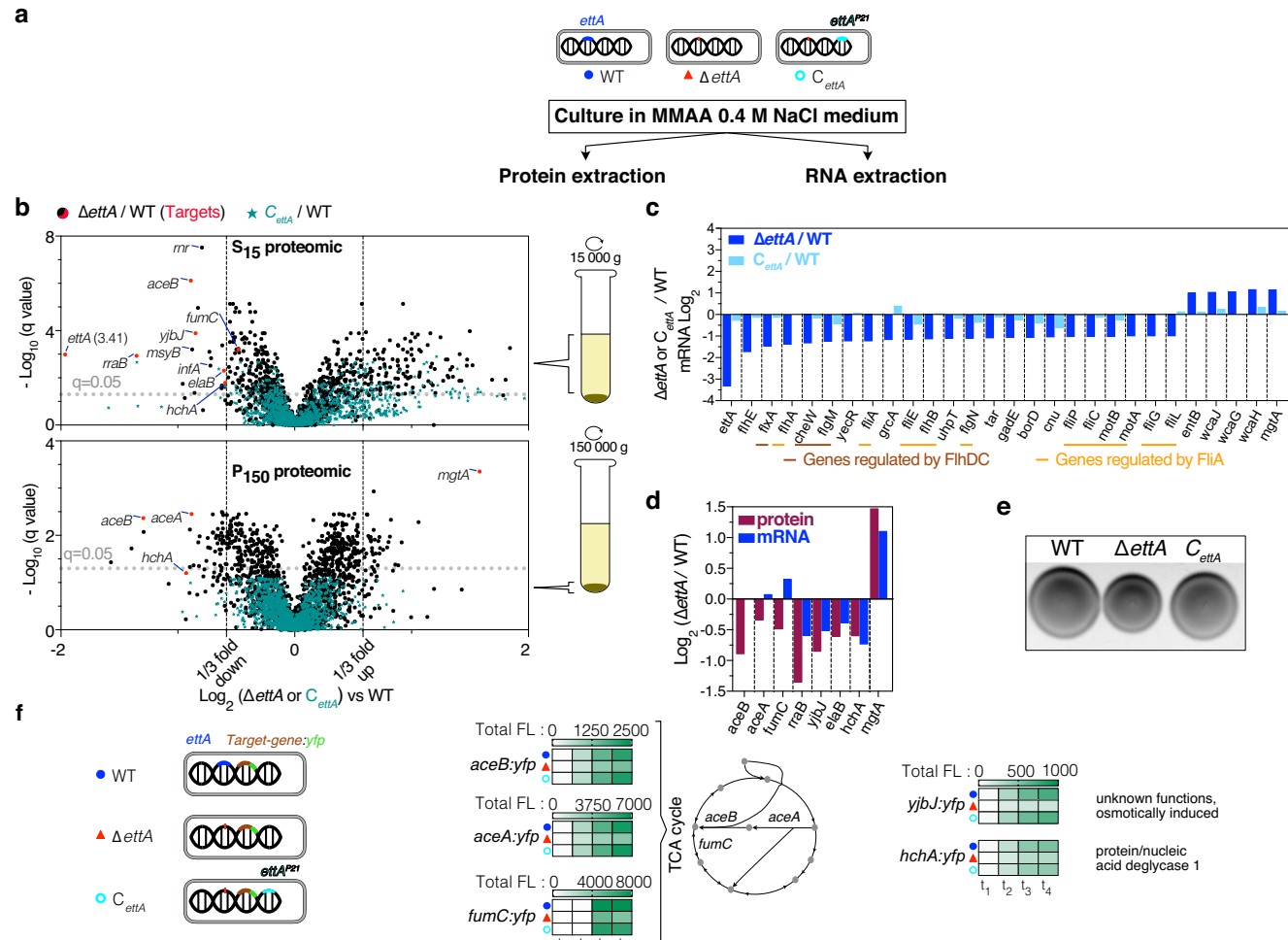

**Fig. 2 | Proteomic and transcriptomic analysis of ΔettA vs. WT strains reveal post-transcriptional downregulation of *aceB, aceA, fumC, yjbJ*, and *hchA* genes expression. a** The WT, ΔettA, and C_ettA strains were cultivated in MMAA medium containing 0.4 M NaCl. Cells were harvested, and proteins were extracted into two fractions: S_15 (proteins in the soluble extract) and P_150 (proteins in the pellet after centrifugation at 150,000 × *g*) along with total RNA. **b** Volcano plots depict proteomic analysis of the fractions S_15 and P_150 fractions. The ratios ΔettA/WT (black) or C_ettA/WT (green) of protein quantity and their corresponding *q*-value (corrected *p*-value for false discovery rate using Benjamini–Hochberg test) are presented in log_2 and -log_10 scales, respectively. Proteins discussed in the text are named. Red dots indicate the proteins downregulated in the ΔettA strain that were further studied. **c** Bar graph illustrating log_2 ratios of some deregulated genes in ΔettA

strain from transcriptomic analysis (ΔettA/WT in blue bars; C_ettA/WT in light blue bars). Brown-underlined genes are regulated by FlhDC while orange-underlined genes are regulated by FliA. **d** Bar graph displaying log_2 ratios of ΔettA / WT in proteomic (burgundy red bars) and in transcriptomic (blue bars) of deregulated genes in ΔettA strain that have been validated independently. **e** Motility tests of the three strains (WT, ΔettA, and C_ettA) on 0.3% agar plates containing MMAA medium, showing a reduced motility for the ΔettA strain. **f** Heatmaps of the YFP total fluorescence level of the same three strains with a *yfp* translational fusion inserted in the genome in frame with the *aceB, aceA, fumC, yjbJ*, and *hchA* genes (*aceB:yfp, aceA:yfp, fumC:yfp, yjbJ:yfp* and *hchA:yfp*) grown in MMAA medium or LB medium for *fumC:yfp*.

we tested media with single TCA intermediates as carbon sources. This screen (Fig. 1c and Supplementary Fig. 1d) shows that the growth inhibition by salt in the ΔettA strain occurs specifically when malate, fumarate, and succinate were used. To our surprise, growth on glyoxylate produced a reversed phenotype where ΔettA grows faster than the other strains.

Proteomic studies were performed on the supernatant (S_15) or pellet (P_150) fractions from protein extracts from cultures grown in MMAA-NaCl medium (Fig. 2a). These studies identified several differences in protein expression in the ΔettA strain compared to the WT or the C_ettA strains (Fig. 2b). By coupling it with transcriptomic studies (Fig. 2c, d) on the same cell extract for the S_15 sample, we compared the change of expression in the ΔettA strain at the protein or mRNA level (Fig. 2d). For *aceB, aceA*, and *fumC* genes, the corresponding mRNAs were expressed at the same level as the WT suggesting a post-transcriptional regulation (Fig. 2d). For *rraB, yjbJ, elaB*, and *hchA* genes, both the mRNA and protein levels decreased. Since the mRNA decay

can be related to translation because the translation efficiency of an mRNA can alter its stability[50–53], we kept these genes for further analysis. For other genes, a dominant positive effect on the transcription was observed as, for example, *mgtA*, for which the mechanism of regulation by EttA will be described below. A downregulation, primarily at the transcriptional level, was observed for most of the flagella apparatus genes in the ΔettA strain (Fig. 2c). We validated this observation by showing that the ΔettA strain had a motility defect (Fig. 2e).

To confirm the decrease in expression of the five genes (*aceB/A, fumC, yjbJ*, and *hchA*) in the ΔettA strain, we constructed *yfp* reporters, where the *yfp* gene, lacking its initiation codon, was inserted immediately prior to the stop codon of the target genes. All the reporters had a lower fluorescence level in the ΔettA strain (Fig. 2f and Supplementary Fig. 2a) which correlated with lower protein expression as confirmed by western blot (Supplementary Fig. 2b). For *aceB/A* and *fumC* there was no change in their transcription level as determined by

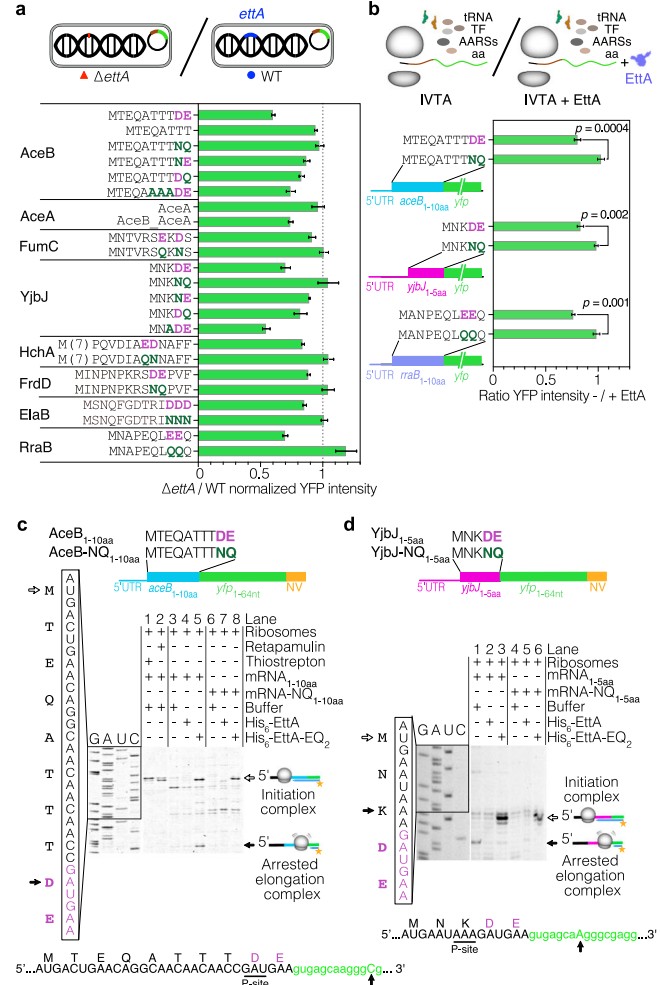

**Fig. 3 | EttA assists ribosomes during the synthesis of peptides with repeated acidic residues. a** The beginning of the coding sequences of the *aceB* (malate synthase), *aceA* (isocitrate lyase), *fumC* (fumarase C), *yjbJ* (putative stress-response protein), *hchA* (protein/nucleic acid deglycase 1), *frdD* (fumarate reductase subunit D), *elaB* (tail-anchored inner membrane protein) and *rraB* (ribonuclease E inhibitor protein B) genes were cloned in a low-copy plasmid (pMMB) under the control of an IPTG inducible promoter and translationally fused to a *yfp* gene. The corresponding aa sequence is presented on the left side. Point mutations tested (dark green) of the acidic residues (purple) responsible for the EttA-dependent expression are indicated in the sequence, other mutations upstream of the acidic residues were also tested for *aceB* and *yjbJ*. Two *aceA:yfp* fusions were made, one with only the intergenic *aceB-A* sequence and the second one with the operon sequence (*aceB* gene and the intergenic region). Cultures were performed in LB_Amp100 medium in the presence of 1 mM IPTG, to avoid differential growth between the strains. Histograms show the ratios of YFP intensity in the ΔettA vs. WT strain for the different constructs (details of the calculation of the ratio and propagation of the error are given in the "Methods" section) for the different constructs. Ratios are measured at the end of growth. The growth curves, results for the complemented strain (C*ettA*), and details of the constructions are presented in Supplementary Fig. 3. **b** Histograms showing the ratios of YFP intensity in the presence or absence of 5 μM His6-EttA after 150 min of in vitro translation (NEB PURExpress) of the different transcripts (*aceB*1-10aa*:yfp*, *yjbJ*1-5aa*:yfp* and *rraB*1-10aa*:yfp*) and for the same ones harboring the mutations that abolish EttA dependency in vivo. The *p*-values were determined by unpaired two-tailed *t*-tests. **a**, **b** Error bars represent mean ± s.d. for triplicate experiments. **c** Toeprinting assay using *aceB*1-10aa and *aceB*-NQ mRNA with or without His6-EttA or His6-EttA-EQ2 at 5 μM. The position of ribosomes stalling at the initiation codon was verified by the addition of Thiostrepton and Retapamulin. **d** Toeprinting assay of *yjbJ*1-5aa and *yjbJ*-NQ mRNA with or without His6-EttA or His6-EttA-EQ2 at 5 μM. **c**, **d** The construct of the mRNA used for toeprinting is indicated above the gels. Empty and plain arrows indicate ribosomes blocked at the initiation or arrested on a specific motif during elongation, respectively. Below the gel, the sequence with the toeprint signal for the stalling (black arrow) and the possible codon corresponding P-site codon (underlined, 15–16 nt away from the toeprinting) is indicated. The experiments have been reproduced at least twice.

northern blot, but for *yjbJ* and *hchA* transcript, levels decreased in the ΔettA strain. Other related genes of the metabolic pathway were tested, but they either did not show any change between the strains (*maeB*, *fumB*, *frdB*) or the fluorescence levels were too low to be quantified (*frdA/C/D*, *glcB*, and *elaB*) (Supplementary Fig. 2a).

## EttA alleviates ribosome pausing on acidic residues

In a second approach to disengage the five validated *yfp*-reporter fusions from their genomic transcriptional environment, we inserted their sequences from the 5′ untranslated region (UTR) of target genes up to the *yfp* stop codon in a low-copy plasmid (pMMB) under control of an IPTG inducible promoter. The constructs for *aceB*, *fumC*, *hchA*, and *yjbJ*, which showed lower expression in the ΔettA strain from their own promoter, also showed lower expression after IPTG induction of the pMMB plasmid in the ΔettA compared to the WT or the C*ettA* strains. However, constructions harboring only the 5′UTR up to the initiation codon of *aceB* or *yjbJ* were expressed independently of EttA (Supplementary Fig. 3a, b) showing that the translated sequence is necessary for regulation by EttA.

To determine the minimal sequence required for the EttA-dependent expression change, we conducted truncations within the target genes (Fig. 3a and Supplementary Fig. 3a, b). Constructs with the first 10 or 60 residues of AceB remained subjected to EttA regulation, whereas the construct with only the first eight residues lost this regulation. The absence of Asp9 and Glu10, two acidic residues, in this construct, lead to the loss of differential expression in the ΔettA strain. Additionally, replacing one acidic residue by a neutral one (Asn for Asp or Gln for Glu) reduced the EttA-dependent effect, while replacing both completely abolished it (Fig. 3a and Supplementary Fig. 3a, b).

The replacement of the three threonines preceding the acidic residues, by alanine did not affect much the regulation confirming the importance of the acidic residues in the EttA-dependent regulation. For the gene *aceA* which is in an operon after the gene *aceB*, regulation by EttA was observed only when *aceA:yfp* was expressed as part of the *aceB-A* operon, implying a co-translation of the two genes (Fig. 3a and Supplementary Fig. 3a, b), with the translation of *aceA* being dependent of the rate of translation of *aceB*.

Similarly, for *fumC*, *yjbJ*, and *hchA* genes, the EttA-driven regulation was also maintained in constructs that contain at least the first 5–20 N-terminal residues of the encoded protein sequences (FumC: 10 aa, YjbJ: 5 aa and HchA: 20 aa). For these three genes, the replacement of two acidic residues (Glu7 Asp9 for FumC, Asp4 Glu5 for YjbJ, and Glu15 Asp16 for HchA) by neutral residues abolished the regulation by EttA (Fig. 3a and Supplementary Fig. 3a, b). Replacement of only one acidic aa in YjbJ as for AceB reduced the EttA-dependent effect. Overall, these results strongly suggest that EttA can facilitate the translation of mRNA encoding acidic residue repeats at the beginning of their sequence.

In light of these findings, we revisited a subset of our early proteomic targets, some of which we had not previously examined or could not detect the expression with our YFP fusion reporters (Supplementary Fig. 2a). We noticed that FrdD, ElaB and RraB contain two or three acidic residues early in their sequences (Asp10 Glu11 for FrdD, Asp11 Asp12 Asp13 for ElaB and Glu8 Glu9 for RraB). We applied the same plasmid-based expression approach to those targets and showed that YFP reporters containing the early part of the target's sequences (FrdD 14 aa, ElaB 15 aa, and RraB 10 aa) exhibited a dependency on EttA for their expression. Moreover, the replacement of the acidic residues by neutral ones eliminated the EttA dependency (Fig. 3a and Supplementary Fig. 3a, b).

To demonstrate the translational regulation driven by EttA, we performed in vitro translation assays (IVTAs) using T7-transcribed mRNA from the shortest YFP constructs that exhibited EttA-dependent expression changes ($aceB_{1-10aa}$, $yjbJ_{1-5aa}$ and $rraB_{1-10aa}$) along with their respective mutants that alleviated this effect ($aceB$-NQ$_{1-10aa}$, $yjbJ$-NQ$_{1-5aa}$ and $rraB$-QQ$_{1-10aa}$). IVTA of all the non-mutated mRNAs showed an increase of expression when purified and partially monomeric His$_6$-EttA protein (Supplementary Fig. 4a) was added to the reaction in comparison to the reaction without EttA (Fig. 3b). However, addition of His$_6$-EttA had no effect on mRNAs where the acidic amino acids had been replaced by neutral residues (Fig. 3b and Supplementary Fig. 4b).

Then, we performed some toeprinting assays to investigate if the acidic residues produce a stalling of the ribosome during translation of the mRNA. First, we used retapamulin and thiostrepton, two antibiotics that stall ribosomes at start codons, to confirm the translational start site (Fig. 3c). The toeprint of the $aceB_{1-10aa}$ and the $aceB$-NQ$_{1-10aa}$ constructs were similar in the absence of EttA. The addition of EttA had no effect, but EttA-EQ$_2$ produced two stalling events. The first one at the initiation site on both $aceB_{1-10aa}$ and the $aceB$-NQ$_{1-10aa}$ constructs is consistent with previous results demonstrating that EttA-EQ$_2$ binds to ribosomes with a free E site, *i.e.* primarily initiating ribosomes[2,3,14,15]. The second stalling event is only present in the $aceB_{1-10aa}$ construct and corresponds to the Asp9 codon in the ribosomal P-site. In contrast, $yjbJ$ toeprinting (Fig. 3d) showed a specific stalling on WT $yjbJ_{1-5aa}$ in addition to the toeprint band of the initiation complex (determined here by toeprinting in the presence of EttA-EQ$_2$ which induced stalling at initiation). This second toeprint signal was not observed when WT EttA was added but was still faintly detected with EttA-EQ$_2$. This toeprint signal corresponds to the Lys3 in the P-site of the ribosome. We tested if Lys3 had an effect on the regulation by EttA by replacing it by an Ala in our $yjbJ_{1-5aa}$·$yfp$ reporter. The related construct showed a similar decrease in expression in the absence of EttA as the WT construct in vivo, demonstrating that Lys3 had no impact on the EttA-dependent regulation (Fig. 3a and Supplementary Fig. 3a, b), therefore it is possible that in this case, the toeprinting signal does not reflect the true positioning of the P-site tRNA possibly due to alternative ribosome conformation.

Altogether these results argue that polymerization of acidic residues triggers ribosome pausing, leading to two potential outcomes. Firstly, in the case of $aceB$, the absence of a toeprint signal in the absence of EttA, juxtaposed with its presence in the presence of His$_6$-EttA-EQ$_2$, strongly suggests ribosome dissociation that can be stabilized by EttA-EQ$_2$. This possible dissociation is further corroborated by a recent preprint in which the authors show that the introduction of three Glu residues at the beginning of the LacZ protein sequence triggers ribosome dissociation during in vitro synthesis, as evidenced by the formation of a peptidyl-tRNA, which can be rescued by EttA[54]. Secondly, for $yjbJ$, a distinct ribosome stalling event is observed, characterized by the detectable toeprinting signal. Notably, EttA is consistently demonstrated as an effective modulator, successfully mitigating ribosomal pausing and thereby augmenting protein expression in all instances. These observations agree with recent reports showing that repeated acidic residues can lead to intrinsic ribosome destabilization[55,56].

## EttA-dependent regulation of aceA and aceB expression modulates the physiology of the central metabolism

Three of the proteins for which the synthesis is directly regulated by EttA: AceB (malate synthase), FumC (fumarase C), and FrdD (fumarate reductase subunit D), are enzymes of TCA and glyoxylate shunt pathways. More specifically, they catalyze reactions that use the substrates for which we identified a growth phenotype for the Δ$ettA$ strain (Fig. 1c). A fourth enzyme, AceA (isocitrate lyase) is indirectly regulated by EttA through the regulation of $aceB$, the upstream gene of the operon (Fig. 3a). AceA catalyzes the conversion of isocitrate into succinate and glyoxylate while AceB catalyzes the conversion of glyoxylate and acetyl-CoA into malate and CoA. Together they form the glyoxylate shunt, a metabolic pathway found in some bacteria and plants that bypasses the carbon dioxide-producing steps of the TCA cycle (Fig. 1c).

It is often assumed that glyoxylate in *E. coli* is metabolized into malate by AceB. However, early publications have demonstrated that the optimal route for glyoxylate metabolism involves glyoxylate carboligase[57,58] (Gcl), which condenses two glyoxylates into tartronate semialdehyde. This compound is then incorporated into the glycolysis pathway through its conversion to glycerate and subsequently to 2-phosphoglycerate[59,60] (Fig. 4a). Accordingly, a deletion of the $gcl$ gene is expected to have a drastic effect on *E. coli* growth on glyoxylate and indeed abolished the growth of the three strains on MM with this carbon source (Fig. 4b and Supplementary Fig. 5a). We hypothesized that the repression of AceB expression (35% less in Δ$ettA$ strain) promotes the metabolic flux toward the $gcl$ pathway, thus increasing the growth of the Δ$ettA$ strain on glyoxylate (Fig. 4b, bottom). If so, deletion of $aceB$ should increase growth on glyoxylate for all the strains and overexpression should produce the reverse effect. This prediction was confirmed experimentally, where the deletion of $aceB$ increased the growth and all the strains grew at the same rate (Fig. 4b and Supplementary Fig. 5a). Overexpression of $aceB$ reduced the growth rate for all the strains, but the Δ$ettA$ strain retained a small advantage of growth. Overexpression of the AceB-NQ variant, for which EttA had no effect on its expression (Fig. 3a), also reduced the growth rate of all the strains, but equally in all three strains (Fig. 4b and Supplementary Fig. 5a), as expected if the phenotype was due to the regulation of AceB by EttA. The deletion of the second malate synthase gene ($glcB$), which is expressed at low level in our experimental conditions (Supplementary Fig. 2a), also improved growth of all the strains on glyoxylate, but the Δ$ettA$ strain maintained a small advantage (Fig. 4b and Supplementary Fig. 5a). As expected, the deletion of $aceA$ gene had no effect on this phenotype.

To understand the metabolic stress induced by EttA deletion in the MMAA-NaCl medium, we tested the growth of the strains deleted for genes encoding TCA enzymes in this medium. Notably, $aceA$ was essential for *E. coli* growth on MMAA medium with 0.4 M NaCl (Fig. 4c and Supplementary Fig. 5b) but not in the same medium without salt (Supplementary Fig. 5c). These results demonstrate the importance of $aceA$, which catalyzes the first step of the glyoxylate shunt, during salt stress. Although the glyoxylate shunt's implication in desiccation and salt stress tolerance in eukaryotic organisms has been established[61,62], its importance in *E. coli* for salt tolerance has not been previously reported. The gene $aceA$ is less expressed in the Δ$ettA$ strain due to a polar effect of the translation regulation of $aceB$ by EttA (Fig. 3a). Therefore, since $aceA$ is important for the growth in the salt medium, the lower expression of $aceA$ in the Δ$ettA$ strain might account for the growth defect of this strain in MMAA-NaCl medium. Consistent with this hypothesis, ectopic overexpression of $aceA$ (which is no more regulated indirectly by EttA) reduced the growth discrepancy between the three strains (Fig. 4c and Supplementary Fig. 5b). The deletion of $aceB$ in this medium slightly increased the growth differences (Fig. 4c and Supplementary Fig. 5b). Deletion of $gcl$ severely inhibited the growth of the WT and C$_{ettA}$ strains specifically in presence of NaCl but had no strong effect on the Δ$ettA$ strain (Fig. 4c and Supplementary Fig. 5b, c). These observations suggest that the WT strain uses the conversion of glyoxylate to tartronate semialdehyde in the high salt condition more than the Δ$ettA$ strain. The correlative of that is that the Δ$ettA$ strain uses the glyoxylate shunt pathway less and therefore likely funnels more metabolic flux through the step of the TCA cycle catalyzed by the isocitrate dehydrogenase encoded by $icd$ (Fig. 4c, bottom). Accordingly, deletion of the $icd$ gene, which prevents the metabolic flux through the carbon dioxide-producing steps of the TCA cycle, increased the salt sensibility of the Δ$ettA$ strain as did the

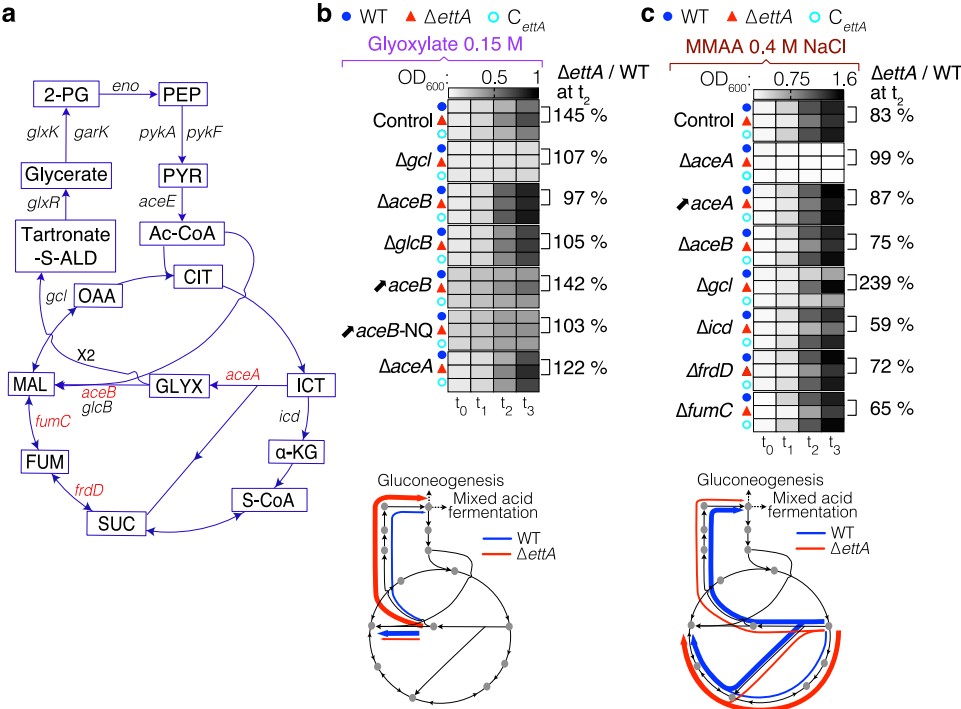

**Fig. 4 | EttA-dependent synthesis of some proteins of the TCA/glyoxylate shunt pathways can explain the observed phenotypes of the Δ*ettA* strain.**
**a** Representation of the TCA cycle, the glyoxylate shunt, and the tartronate-semialdehyde pathway showing intermediary products and genes encoding the enzymes. The expression of genes shown in red is dependent on EttA. **b, c** Top: Heatmaps show the $OD_{600}$ at various times of growth as in Fig. 1a from cultures in MM.Glyoxylate (**b**) or MMAA 0.4 M NaCl (**c**) media, on the right of the heatmap the percentage of the ratio Δ*ettA*/WT is indicated for the $t_2$ time point. The control experiments for the WT, Δ*ettA*, and C*ettA* are presented on the top. The other

heatmaps are for the same strains with a deletion of either *aceB, aceA, glcB, gcl, fumC, frdD*, or *icd* gene or for the same strains overexpressing *aceB* or *aceB*-NQ from a plasmid (↗*aceB* and ↗*aceB*-NQ). For *aceA* overexpression (↗*aceA*) from a plasmid, the three strains deleted of *aceA* were used. The OD equivalents of the heatmap color shadings are indicated above the control experiments (WT, Δ*ettA*, and C*ettA*) for the two tested media. Growth curves and time points used for the heatmaps are presented in Supplementary Fig. 5. Bottom: Representation of the possible metabolic flux in MM.Glyoxylate medium (**b**) and MMAA 0.4 M NaCl medium (**c**) for the WT (blue) and Δ*ettA* (red) strains.

deletion of *fumC* and *frdD* (Fig. 4c and Supplementary Fig. 5b), two genes that are less expressed in this strain. Overall, these results suggest that the growth defect of the Δ*ettA* strain is partly due to the lower expression of *aceA* which forces the bacteria to rely more on the carbon dioxide-producing steps of the TCA cycle. The WT strain uses more the glyoxylate to tartronate-semialdehyde pathway that is more beneficial metabolically under salt stress because this route reduces respiration by funneling the metabolic flux towards mixed acid fermentation and/or by increasing synthesis of osmoprotectants by the gluconeogenesis pathway (Fig. 4c bottom).

**Change of *mgtA* expression is mediated by the action of EttA on the synthesis of the leader peptide MgtL**
The proteomic study on the $P_{150}$ protein extracts, which included membrane proteins, showed a 3-fold increase in the Δ*ettA* strain of the protein MgtA, an ATP-dependent $Mg^{2+}$ importer[63]. The expression of the gene *mgtA* is under the regulation of the leader peptide MgtL[55,64,65]. During cellular growth in conditions where $Mg^{2+}$ is not limited, translation of *mgtL* by the ribosome maintains a Rho-dependent terminator (RDT)[55,64,65] accessible to terminate the transcription of the mRNA and therefore prevents the transcription of *mgtA* (Fig. 5a). When $Mg^{2+}$ becomes limiting, the low intracellular concentration favors ribosome dissociation during the translation of *mgtL*. This ribosomal drop-off occurs during the translation of the destabilizing motif EPDP and leads to a conformational change of the mRNA that prevents accessibility to the RDT and transcription termination[55], therefore *mgtA* is transcribed (Fig. 5a).

To test if EttA could modulate the translation of *mgtL* and consequently the transcription of *mgtA*, we constructed two reporters

using the same plasmid, one with the sequence of *mgtL* (from the 5′ UTR to the codon before the stop codon) fused to the *yfp* and the other one from the same 5′UTR extended to the 5th codon of *mgtA* fused to the *yfp* (Fig. 5b and Supplementary Fig. 6a, b). When strains were grown in the MMAA medium with 2 mM or 0.05 mM of $Mg^{2+}$, the fluorescence signal was lower in the Δ*ettA* strain for the *mgtL* fusion and higher for the $mgtA_{1-5aa}$ fusion, in accordance with an action of EttA on the synthesis of the leader peptide MgtL. Northern-blot analysis show that $mgtA_{1-5aa}$ transcript increased in the Δ*ettA* strain (Supplementary Fig. 6c), consistent with the previously reported transcription termination regulation driven by *mgtL* translation[55,65]. Mutation of Glu2 and Asp4 of the destabilizing EPDP motif to Gln and Asn, respectively, resulted in an increased of the expression of the *mgtL* fusion in the Δ*ettA* strain to reach the same level as the WT strain, while, as expected, no detectable expression was then observed for the $mgtA_{1-5aa}$ fusion (Fig. 5b and Supplementary Fig. 6a, b).

We used IVTA to determine if EttA directly regulated *mgtL* expression (Fig. 5c and Supplementary Fig. 6d) These assays showed that the translation of *mgtL yfp* fusion mRNA increased in the presence of EttA, but not for the *mgtL* variant where the acidic residues were mutated. Toeprinting performed on the same constructs showed specific toeprint signal for the WT construct in the presence of $His_6$-EttA a stronger one in the presence of $His_6$-EttA-$EQ_2$ (Fig. 5d). This toeprint signal corresponds to a ribosome stalled with Pro3 in the ribosomal P-site. From these observations, we conclude that EttA can stabilize the ribosome during the synthesis of the destabilizing EPDP motif of MgtL to prevent the drop-off and therefore repress *mgtA* transcription.

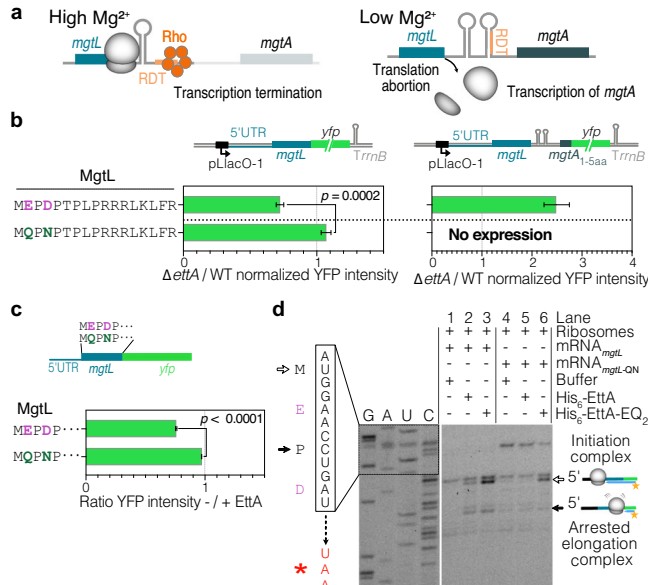

**Fig. 5 | Synthesis of the leader peptide MgtL by the ribosome is dependent on EttA. a** Model of the regulation of *mgtA* expression by Mg²⁺. At high intracellular Mg²⁺ concentrations, the translation of the entire leader peptide MgtL, maintains the mRNA in a conformation where a Rho-dependent terminator (RDT) site is accessible for Rho to terminate the transcription of the mRNA. At low Mg²⁺ intracellular concentrations, dissociation of the elongating ribosome when polymerizing negatively charged residues of MgtL, changes the mRNA into a conformation where the RDT is no longer accessible, allowing the transcription of *mgtA*. **b** Histograms showing the ratios of YFP fluorescence of Δ*ettA*/WT strains, for *mgtL:yfp* fusion (left), or *mgtL_mgtA*₁₋₅ₐₐ:*yfp* fusion (right). Constructs express either the WT *mgtL* or a mutant (*mgtL*-QN) where two acidic residues (purple) were mutated (dark green). The constructions used are shown above the histograms and the sequence of MgtL is on the left. The YFP sequence used in this experiment contains a 6-residues N-terminal translation enhancer[27]. The constructions were expressed from a pMMB plasmid and bacteria were grown in MMAA medium in the presence of Amp100, 1 mM IPTG, and 2 mM MgSO₄. Ratios are measured at the end of growth (see Supplementary Fig. 6b). **c** Design of the mRNA used for IVTA. Histograms showing the ratios of YFP intensity with or without His₆-EttA (5 µM) after 150 min of in vitro translation of mRNA coding for WT or MgtL-QN YFP fusions. **b**, **c** Error bars represent mean ± s.d. for triplicate experiments. Two-tailed *t*-tests were used to measure the *p*-value. **d** Toeprinting assay of *mgtL* and *mgtL*-NQ mRNA with or without His₆-EttA or His₆-EttA-EQ₂ at 5 µM. Empty and plain arrows indicate ribosomes blocked at the initiation or arrested on a specific motif during elongation, respectively. This experiment has been reproduced at least twice.

## The acidic residues need to be within the 30 first aa of the sequence for protein synthesis to be EttA-dependent

We have identified eight proteins for which the synthesis is dependent on EttA and for all of them, acidic residues are necessary for this effect. Consequently, we aimed to create a motif derived from these eight sequences, which includes the acidic residues and a few neighboring ones. The designed motif NKDEPD, is not naturally found in the *E. coli* proteome. We inserted this motif sequence after initiating methionine of the YFP sequence in two different constructs: one with the 5′UTR of *aceB* and the other one with the 5′UTR of *yjbJ*. Both constructs had a strong EttA dependency for their expression (Fig. 6a and Supplementary Fig. 7a, b), confirming that we can design an artificial sequence that relies on EttA for optimal expression. By reproducing the effect in vitro in an IVTA using the 5′UTR *aceB* construct, we confirmed that this occurred by direct interaction of EttA on the translating ribosome (Fig. 6b and Supplementary Fig. 7c). We also tested a construct that expressed the same aa sequence but used different synonymous codons (syn-codons). This construct showed a drop in the overall expression level, but the EttA dependency on its expression

was maintained. Therefore the EttA regulatory effect is dependent on the aa, but not on the codons used (Fig. 6a and Supplementary Fig. 7a, b).

To determine if the position of this sequence had an importance for the EttA-dependent expression, we tested several constructs where this sequence was moved by increments of 10 aa away from the initiating methionine. We used the sequence of AceB-NQ, whose expression is no longer EttA-dependent, to test the 10 aa increments. Displaced by 10 aa, the consensus sequence expression was still strongly induced by EttA. When displaced by 20 aa, there was only a small, not statistically significant, induction by EttA, while for the 30 and 40 aa constructions there was no effect of on their expression (Fig. 6a and Supplementary Fig. 7a, b). This experiment demonstrates that the acidic residues need to be within the first 30 aa of a protein sequence to make its expression EttA-dependent.

Since we found that two acid residues in the beginning of the sequence can make protein synthesis-dependent on EttA, we wondered if it could be detected genome-wide in our proteomic studies. We searched for the presence of a motif where two acidic residues are adjacent or separated by one aa in the first 50 aa of *E. coli* proteome (Motif: −/X₀,₁/−). Then we grouped them according to the position of the acidic residues within segments of 10 aa and compared the ratio of expression of the WT versus the Δ*ettA* or C*ettA* strains (Fig. 6c and Supplementary Fig. 7d). This analysis revealed that the presence of the motif within the first 20 aa correlated with a statistically significant decrease of expression only in the Δ*ettA* strain and with a stronger effect when it is located within the first 10 aa.

The analysis of the proteomic data also revealed that certain proteins containing double acidic residues early in their sequences remain unaffected by *ettA* deletion. Furthermore, the YFP sequence begins with "MVSKGEE" and its expression remains unaffected by EttA (Supplementary Fig. 7e–f). We thus attempted to render YFP expression dependent on EttA by introducing sequential mutations in the acidic residues and their surroundings (Supplementary Fig. 7e–f). Constructions with an EE or DE motif alone had no EttA dependency for their expression. Even though the ED motif showed a statistically significant change, it was only around 7%, which is less than changes observed with EttA's physiological targets. In fact, at a minimum, a DPEE motif was necessary to make the construct expression clearly dependent on EttA. The presence of a Pro suggests that a structural constraint on the positioning of the nascent peptide is crucial for this phenomenon to occur.

## Discussion

We have demonstrated that EttA enhances translation of specific mRNA (*aceB*, *rraB*, *yjbJ*, *fumC*, *frdD*, *hchA*, *elaB* and *mgtL*). This specificity arises from the destabilizing effect of early acidic residues during protein synthesis, which EttA mitigates. According to our proposed model (Fig. 6d), EttA initially binds to a ribosome that has paused at the acidic residues, most likely before the peptide bond formation of the second acidic residue occurs (Figs. 3c and 5d). In this state, it can avert ribosome dissociation, as exemplified by *aceB* and *mgtL* (Figs. 3c and 5d), where the toeprinting didn't detect stalling in the absence of His₆-EttA, but did detect stalling with His₆-EttA-EQ₂, suggesting that the non-hydrolytic EttA mutant can prevent the ribosome dissociation but cannot restore the elongation. In the case of *yjbJ*, the initial pausing does not lead to ribosomal drop-off since a clear stalling signal is seen on the toeprinting experiments in the absence of EttA (Fig. 3d). In the final step, EttA allows the elongation to restart. Ribosomal dissociation during elongation of acidic residues has been previously demonstrated for MgtL and other proteins in prokaryotes[55,56] and eukaryotes[66], this effect was described as Intrinsic Ribosome Destabilization (IRD). Moreover, a recent report suggests that EttA can prevent the IRD in vitro[54].

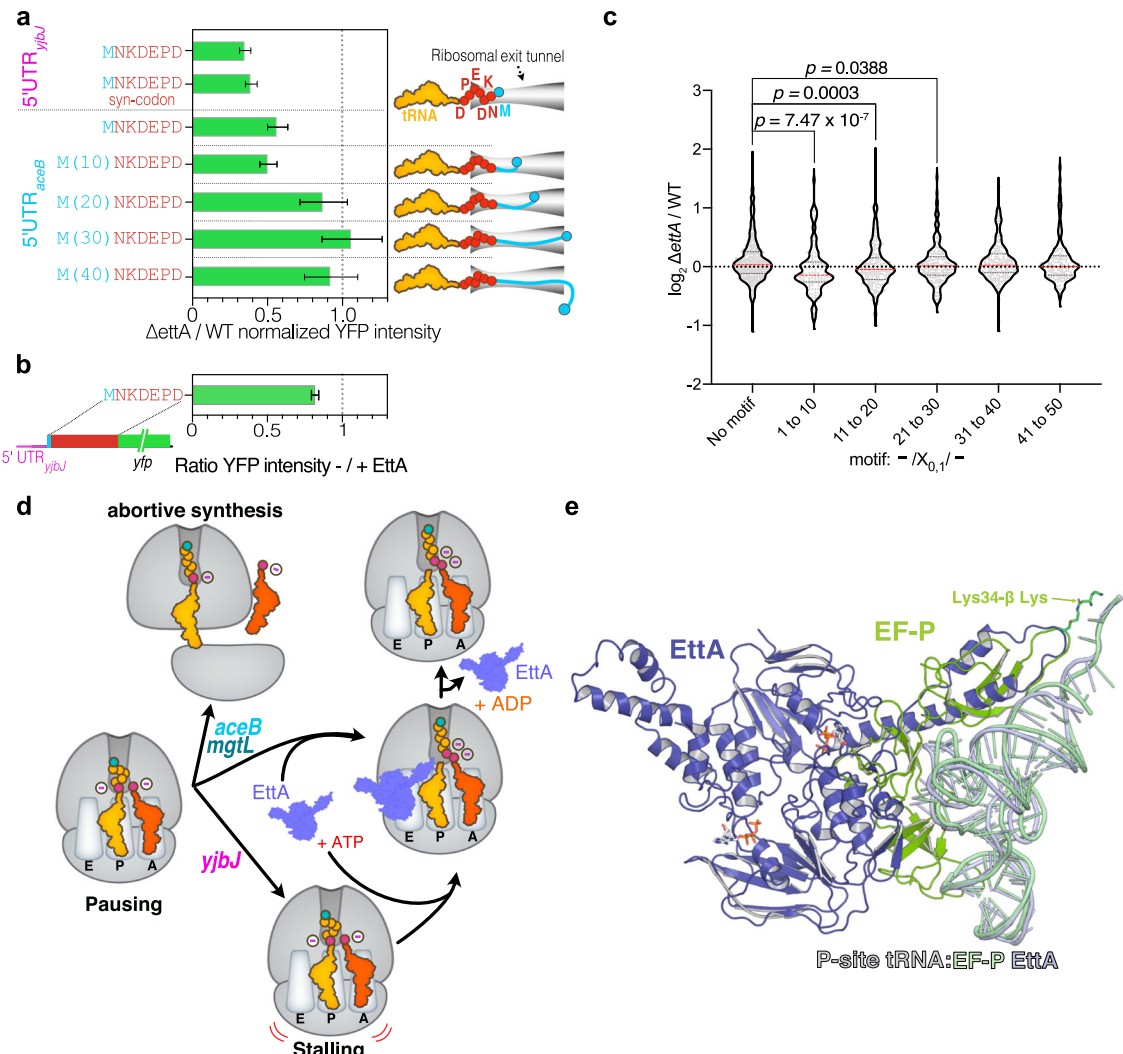

**Fig. 6 | EttA rescues translation of mRNA encoding acidic residues at the beginning of translated sequences and shares a structural homology with EF-P.** **a** Histograms showing the ratio of the YFP fluorescence emission of Δ*ettA*/WT strains expressing the 5′UTR of *yjbJ* followed by a *yfp* gene where the exogenous coding sequence NKDEPD has been inserted after the initiating methionine using two different sets of synonymous codons (syn-codon, see "Methods" section). A similar construct with the 5′UTR of *aceB* is also presented as well as equivalent constructs where segments (10, 20, 30 or 40 aa) of the AceB-NQ sequence have been introduced between the initiating methionine and the NKDEPD sequence. Cartoons of the translated peptides in the exit tunnel of the ribosome are shown on the right. Results for the complemented strain (*C_ettA*) and the growth curves are presented in Supplementary Fig. 7a, b. **b** Histograms showing the ratio of YFP intensity after 150 min of in vitro translation of a UTR_*aceB*_MNKDEPD:*yfp* mRNA with or without His_6-EttA (5 μM). Error bars represent the mean ± s. d. for triplicate experiments for (**a**, **b**). **c** Violin plot showing the expression level of genes containing the ExD motif (−/X_{0,1}/−) or not within the first 50 neo-synthesized aa by windows of 10 aa based on the log_2 Δ*ettA*/WT ratios obtained in the S_15 proteomic.

The plot show genes expression distributions over $n = 488$ for absence of the motif, $n = 121$ for motif between residues 1 and 10, $n = 204$ for motif between residues 11 and 20, $n = 184$ for motif between residues 21 and 30, $n = 204$ for motif between residues 31 and 40 and $n = 152$ for motif between residues 41 and 50. The red and gray dotted lines represent the median and quartiles respectively. The *p*-values are calculated using the two-tailed Mann–Whitney test. The same analysis for the complemented strain (*C_ettA*) is presented in Supplementary Fig. 7d. **d** Model of EttA function to rescue the expression of genes with the DxE motif. When ribosomes encounter repeats of acidic residues (adjacent or separated by one residue) during the early stage of the synthesis of the nascent elongating peptide, they stall and in certain cases, they dissociate from the mRNA (as for *aceB* and *mgtL* genes). EttA can probe stalled ribosomes with an empty E site and restore the elongation process, then it dissociates using its ATPase activity. **e** Structure of EttA-EQ_2 (dark blue) in complex with 70S IC complex (EMD-29398)[18] aligned on the domain V of the 23S rRNA of the structure (6ENJ)[68] of the polyproline-stalled ribosome in complex with EF-P (green).

This model draws parallels with the function of the translation factor EF-P[67], which enhances the translation of polyproline-containing proteins[24,25]. Polyproline-containing nascent chains adopt a conformation that is incompatible with the ribosome's NPET structure, creating tension on the peptidyl-tRNA[68] that prevents the formation of the peptide bond with the Pro-tRNA in the A site. EF-P binding to the stalled ribosome forces the polyproline-containing nascent chain to adopt a correct conformation. While the structures of EF-P and EttA differ significantly (Supplementary Fig. 8a), comparative analysis of their ribosome complexes[18,68] aligned on domain V of 23S rRNA,

highlights the close proximity of the loop in EF-P's extension containing the essential β-lysyl-lysine residue and the loop in EttA's PtIM, with distances ranging from 1.9 to 3 Å (Fig. 6e and Supplementary Fig. 8b).

These structural and functional similarities occur in the region of the proteins that interact with the acceptor stem of the P-site tRNA. Furthermore, the structures of EttA-EQ_2 in complex with initiating or elongating ribosomes[18] indicate that the geometry of the P-site tRNA is conducive to peptide bond formation, akin to the structures observed in EF-P ribosomal complexes[68]. Additionally, a recent preprint

described an ABC-F protein in *Bacillus subtilis* that can partially overlap with the function of EF-P[69].

The stalling caused by polyproline is context-dependent, depending on the position within the sequence and on the sequence context[70,71]. Similarly, the stalling induced by acidic residues and rescued by EttA, is not universal. Although we have shown EttA's influence on several proteins with acidic residue repeats within the first 30 amino acids (Figs. 3, 5b, 6a and Supplementary Fig. 7e), we have also identified other proteins with this feature that do not rely on EttA for their expression (Fig. 6c). Therefore, the mere presence of di-acidic residues is insufficient to destabilize translating ribosomes and render translation EttA-dependent. The conformation of the nascent peptide in the ribosome's NPET appears to be crucial, with the presence of a proline upstream of the acidic residues in four of our eight targets supporting this notion. However, fundamental differences exist between EF-P and EttA. In the case of polyproline, stalling can occur after the nascent peptide has filled the entire ribosomal NPET[24,25,70,71]. In contrast, the stalling induced by acidic residues, rescued by EttA, is limited to a partially filled NPET. EF-P functions by binding to the stalled complex and by inducing rescue[68], while EttA, employs its ATPase activity following binding, to potentially reshape the nascent peptide's positioning within the NPET/PTC[18].

ARE ABC-F proteins provide resistance against antibiotics that target the PTC and the NPET[5,9–15]. Many of these antibiotics exhibit context-dependent action, meaning the nascent peptide's sequence influences their efficacy[72–75]. Therefore, the sequence of amino acids in the NPET holds significance, paralleling the concept presented here. Despite the absence of an inducer (antibiotic) in EttA's case, this suggests that the actions of ARE ABC-F and ABC-F on the ribosome share similarities and may have evolved from a common rescue mechanism.

Our study reveals that EttA directly influences three genes (*aceB, frdD, fumC*) and indirectly affects *aceA*, all functionally interconnected within the TCA and glyoxylate shunt pathways. This highlights EttA's role in coordinating key metabolic pathways by governing the expression of functionally related genes. Importantly, the malate synthase AceB retains its function when the acidic residues are mutated in the AceB-NQ construct (Fig. 4b). However, its expression is no longer dependent on EttA, implying that these residues can act as a regulatory signal. Additional EttA-dependent genes exhibit functional links. The protein/nucleic acid deglycase 1 (HchA) repairs glyoxal- and methylglyoxal-glycated proteins as well as nucleotides[76,77] by releasing glycolate or lactate from the modified amino acid or nucleotide. The glycolate can be converted into glyoxylate by the Glycolate dehydrogenase (GlcDEF)[78] and will benefit from the EttA-dependent expression of *aceB* for its degradation. The function of the protein YjbJ is unknown, but its expression is abundant during the stationary phase and salt stress[79,80] as is the membrane protein, ElaB, which has been associated with resistance to multiple stresses[81]. The protein RraB modulates RNase E (the main RNase of *E. coli*) activity[82–84], adding another layer of regulation by EttA. MgtA is an ATP-dependent Mg$^{2+}$ importer[63] expressed when cells are deprived of Mg$^{2+}$, its expression regulated by EttA through the leader peptide MgtL, assists in Mg$^{2+}$ homeostasis. This example further demonstrates EttA's potential to repress gene expression using a leader peptide, as documented in certain ARE ABC-F cases[13,27].

EttA plays a pivotal role in regulating the interplay between the TCA and glyoxylate shunt pathways through the translational regulation of *aceB, fumC,* and *frdD* genes. Reduced expression of these genes, particularly *aceB*, and indirectly *aceA* in the Δ*ettA* strain, results in distinct growth phenotypes by reducing the glyoxylate shunt metabolic flux in favor of the TCA cycle or the tartronate-semialdehyde flux. The Δ*ettA* strain exhibits improved growth when glyoxylate is the sole carbon source but performs worse when amino acids are the only carbon source and when subjected to salt stress (Fig. 1a, c). In the MMAA-NaCl medium, the WT strain efficiently manages salt stress by

redirecting metabolic flux through the glyoxylate and tartronate-semialdehyde pathways (Fig. 4c), providing benefits such as increased glycolysis for osmoprotectant production[85,86] (e.g., sucrose and trehalose) and reduced NADH and NADPH production in the TCA carbon dioxide-producing steps. These results support recent studies that have shown the importance of the glyoxylate shunt in stress conditions[61,62,87–89].

These findings underscore EttA's crucial role in the cellular stress-response. Induced by high salt (Fig. 1b) and part of the RpoS regulon[47,48] it may control gene expression during stress through translational regulation. Moreover, our discovery that EttA primarily governs the expression of genes intricately linked to central metabolism, influencing cellular energy levels, suggests that EttA possesses the ability to adapt gene expression according to the prevailing cellular energy state. This adaptation is related to EttA's established dependence on the ADP:ATP ratio[2,3], providing a direct mechanism to connect its regulatory effects to cellular energy status, which warrants further investigation in future studies.

## Methods

### Bacterial strains

The strains used in this study are listed in Supplementary Tables 1–3. The WT reference strain used was the sequenced MG1655[90] strain. The Δ*ettA* strain from a previous study[2] was complemented by the insertion of the *ettA* gene with is native promoter at the P21 genomic locus (strain C$_{ettA}$) or at the *attB* locus (strain C'$_{ettA}$, this strain was used for the Δ*icd* constructions because the P21 locus was not compatible with this deletion). The insertion comprises the ORF of EttA with 105 base pairs upstream and downstream. Insertions in the genome were conducted using the pOSIP clonetegration recombination method[91]. Primers used for the amplification and control of the insertion are described in Supplementary Table 5. Strains harboring a *yfp* fusion with the *aceA, aceB, fumA, fumB, fumC, frdA, frdB, frdD, mqo,* and *glcB* genes were constructed by P1 transduction using a collection of YFP fusions created by the Mori laboratory[92] as donor strains and the three receiver strains MG1655 (WT), Δ*ettA* and C$_{ettA}$. Strains harboring a *yfp* fusion with the *yjbJ, hchA,* and *elaB* genes were constructed by amplifying the *yfp* and chloramphenicol resistance cassettes from the Mori collection[92] with primers containing in their 5′ extremities 60 base pairs homologous to the upstream and downstream regions of the stop codon of the target gene (Supplementary Table 5). Then, the PCR products were introduced by electroporation into MG1655 cells harboring the pKD46 plasmid expressing the $\lambda_{red}$ recombinase and the Gam proteins[93] followed by a selection of chloramphenicol-resistant fusion clones. The constructs were finally transduced with P1 phage into the receiver strains (WT, Δ*ettA,* and C$_{ettA}$). For the strains individually deleted of EttA target genes, deletions from donor strains of the Keio collection[94] were transferred by P1 transduction into the receiver strains (WT, Δ*ettA*, C$_{ettA,}$ and C'$_{ettA}$). All the constructed strains were verified by PCR using the primers described in Supplementary Table 5.

### Plasmids construction

All the plasmids constructed in this study are listed in Supplementary Table 4 and were cloned in the DH5α strain. The pMMB plasmids used for target validation in the WT, Δ*ettA*, and C$_{ettA}$ strains, were derived from the low-copy plasmid pMMBpLlacO-1-67EH. It allows controlled gene expression from the IPTG inducible PLlacO-1 promoter[27] with a transcription start that permits to retain the native 5′UTR of the inserted gene. All of EttA target genes, (except the *mgtL_mgtA*$_{1-5aa}$ insert sequence, which was synthesized by the company TWIST bioscience) were PCR amplified from genomic DNA of the MG1655 strain with primers hybridizing at the transcription initiation site and to the stop codon of the ORF. In the case of multiple transcription start sites we selected the one that generates the shortest 5′ UTR. The primers and the synthetic insert have in their extremities a

homology (15–20 bases) to the PCR products of the pMMB produced for the cloning (Supplementary Table 6). The plasmids were assembled using the NEBuilder HiFi DNA Assembly kit (New England BioLabs). The different truncated or point mutants derived from these plasmids were generated by PCR amplification using primers (Supplementary Table 6) that hybridize at the truncation or contain the point mutations(s) and with 15 bases of homology to allow plasmid recircularization using the NEBuilder HiFi DNA Assembly kit. The pMMB-*aceB* and pMMB-*aceA* plasmids without the YFP fusion were also generated by the same strategy. Plasmid constructs carrying the nucleotide sequence encoding the NKDEPD motif were generated from the plasmid pMMB-*aceB*-NQ:*yfp* by the same approach. The synonymous coding sequence for pMMB-UTR$_{yibL}$ MNKDEPD$_{syn-codon}$:*yfp* changes the original sequence ATGAATAAAGATGAACCAGAT to ATGAATAAA-GACGAGCCAGAC where the three codons encoding acidic residues were replaced by synonymous codons (codons encoding the same residues). All the constructed plasmids were confirmed by sequencing. Most of the constructions with a *yfp* gene express the native YFP venus except for the *mgtL* and *mgtA*$_{1-5aa}$ constructions which express an optimized YFP[27] and for the *yfp* mutant presented in Supplementary Fig. 7a where the first 6 codons systematically use the synonymous codon with the highest number of adenosine[95].

## Growth media and culture conditions

Overnight cultures of the various MG1655 strains were carried out in Luria-Bertani Miller broth (LB-Difco) at 37 °C with agitation. For strains transformed with pMMB or pBAD plasmids, the growth medium was supplemented with ampicillin at 100 μg·ml$^{-1}$ (LB_Amp100) or kanamycin at 50 μg·ml$^{-1}$ (LB_Kan50) and, when required, 0.4% (w/v) of ß-D-glucose was added to repress expression of pBAD plasmids (pBAD-*His*$_6$-*ettA*-EQ$_2$). Bacteria were cultivated in the different media indicated in the figures. A modified MJ9 medium[96], composed of KH$_2$PO$_4$ at 16.5 mM, K$_2$HPO$_4$ at 8.5 mM, NH$_4$SO$_4$ at 15 mM, MgSO$_4$ at 2 mM, MOPS at 133 mM, Tricine at 13.3 mM, supplemented by a 100× dilution of a vitamins mix (MEM VITAMINS, Sigma–Aldrich) and trace elements as described in the original MJ9 medium[96], was used for the cultures. For the MMAA, all the amino acids (Sigma–Aldrich) with the exception of cysteine, tyrosine, tryptophan, and methionine (due to their low solubility) were added at 2 mg·ml$^{-1}$ as a carbon source for a final concentration of glycine at 26.6 mM, alanine at 22.4 mM, valine at 17 mM, leucine at 15.2 mM, isoleucine at 15.2 mM, proline at 17.3 mM, phenylalanine at 12.1 mM, serine at 19 mM, threonine at 16.8 mM, asparagine at 15.1 mM, glutamine 13.7 mM, aspartic acid at 15 mM, glutamic acid at 13.6 mM, arginine at 11.4 mM, histidine at 12.8 mM and lysine at 13.6 mM. Other carbon sources tested were: L-malic acid, sodium fumarate dibasic, succinic acid and pyruvic acid and α-ketoglutaric acid at 0.3 M, glyoxylic acid (MM.Glyoxylate) and oxaloacetic acid at 0.15 M and glucose at 22 mM (MMGlc) all in the MM medium (carbon sources are purchased from Sigma–Aldrich with the exception of α-ketoglutaric acid purchased from Bio Basic). For the assay with salt, NaCl was added at the indicated final concentrations in the figures. Expression of the inserted genes or YFP fusions was induced by addition to the medium of 1 mM of Isopropyl β-D-1-thiogalactopyranoside (IPTG) for the pMMB constructions or 0.2% (w/v) of L-Arabinose for the pBAD constructions.

Cultures in 96-well Falcon microplates were done in triplicate in a volume of 200 μl of medium inoculated with 2 μl of overnight culture (for the MMAA-NaCl test, the cultures were washed with a solution of 0.9% (w/v) NaCl prior to inoculation) and a volume of 60 μl of mineral oil (Sigma–Aldrich) was added to each well in order to prevent evaporation of the medium during growth. Growth of the different strains was followed by measuring the optical density at 600 nm (OD$_{600}$) when incubated at 37 °C under 600 rpm double-orbital shaking with a CLARIOstar plate reader (BMG Labtech). The emission of the YFP fluorescence was recorded at 540–20 nm with an excitation at 497–15 nm. Measurements were taken every 30 min.

## Fitness assays

Competitive fitness assays were conducted as described previously[2] except that different media, described in Supplementary Fig. 1a, were tested (Luria-Bertani (LB) medium from two different providers: Difco Becton Dickinson and USB-affymetrix, MMAA, and MMGlc media described above). Briefly, from overnight cultures in LB-Difco medium of the WT and Δ*ettA* strains, an inoculum composed of an equal ratio of the two strains was used to start a co-culture in the media indicated in Supplementary Fig. 1a. Growth was carried out at 37 °C with shaking for a determined period, then the cultures were sub-cultured in fresh medium (+24 h) to test the viability of the strains during the experiment. PCR assays were performed on cells at the end of each co-culture, using primers hybridizing at ~500 bp upstream and downstream of the *ettA* gene as previously described[2].

## Motility assay

Stationary-phase bacterial cultures were used for the motility test. Overnight precultures of the three strains (WT, Δ*ettA*, and C$_{ettA}$) were washed and diluted to an OD$_{600}$ = 0.5 in physiological water to prepare the inoculum. The motility test was then carried out on a plate containing the MMAA culture medium supplemented with 0.3% (w/v) agar. Plates were inoculated by picking them using a toothpick soaked in an inoculum and then incubated at 37 °C. Representative images of the swimming motility are shown after 48 h.

## Protein extracts and total RNA preparation

Samples for quantification by western and northern blots of protein and mRNA of the *yfp* genomic fusion with *aceA*, *aceB*, *fumC*, *yjbJ*, and *hchA* genes were prepared as following: 96-well microplate culture of each strain (WT, Δ*ettA*, and C$_{ettA}$) with the *yfp* fusion were grown in MMAA or LB medium as described above. For the western blotting, 100 μl of the culture was pelleted, then resuspended in Laemmli (Bio-Rad), in a volume to normalize the OD$_{600}$. For northern blotting, 400 μl of culture were resuspended in 100 μl of RNAsnap buffer (95% (v/v) formamide, 18 mM EDTA, 0.025% (v/v) SDS, 1% (v/v) β-mercaptoethanol) using the RNAsnap method as described previously[97].

## Western blotting

After separation on a 12.5% acrylamide SDS-PAGE gel, proteins were transferred onto a polyvinylidene difluoride (PVDF) membrane for 30 min at 2.5 V and 25 mA using a semi-dry Trans-Blot Turbo system (Bio-Rad). Membranes were blocked by incubating them for two hours at room temperature (RT) in a PBS buffer supplemented with 5% (w/v) of skimmed milk powder and then incubated for 2 h at RT or overnight at 4 °C in 3 ml of PBS 1X + 0.1% (v/v) Tween-20 in presence of a 1000× dilution of anti-GFP antibody (Rabbit anti-GFP, Invitrogen Thermo Fisher Scientific). After washing, the bound antibody was detected after incubation for 1 h in the presence of a secondary antibody in PBS buffer: for *aceB*, *aceA*, and *yjbJ yfp* fusions, a near-infrared fluorescent probe (IRDye 680LT anti-rabbit LI-COR Biosciences) diluted 20,000× in PBS was used whereas for *hchA* and *fumC yfp* fusions, an HRP-conjugated antibody (Anti-rabbit IgG, antibody [HRP] from COVALAB) diluted 20,000× in PBS was used. The HRP-conjugated antibody was detected with the Clarity Western ECL Substrate (Bio-Rad) and signal detection was performed on a Licor Odyssey scanner.

## Northern blotting

For each condition, 6 μg of total RNAs were loaded on a 1% (w/v) agarose gel. Electrophoretic migration was carried out for 75 min at 100 V. RNAs were then transferred to an Amersham Hybond-N+

membrane (Cytiva). Transfer of the RNAs was verified by UV. The RNAs were cross-linked on the membrane by exposing the membrane to UV for 30 s at $100 \times 1200\,\mu J/cm^2$. The radioactive probe was prepared from 40 pmoles of primers described in Supplementary Table 7 and labeled at the 5′ end with 10 units of T4 polynucleotide kinase (New England Biolabs) and $[\gamma\text{-}^{32}P]$-ATP (150 μCi). Membranes were incubated with the probe in the hybridization buffer (ULTRAhyb, Invitrogen Thermo Fisher Scientific) overnight at 42 °C. Membranes were washed three times (once in 2× SSC + 0.1% (v/v) SDS, once in 1× SSC + 0.1% (v/v) SDS, and finally in 0.1× SSC + 0.1% (v/v) SDS) at 42 °C during 15 min. The radioactive signal was detected on a Typhoon 9 500 FLA scanner (GE Healthcare).

### RNA-seq and proteomic experiments

Cultures of the three strains (WT, $\Delta ettA$, and $C_{ettA}$) in a volume of 125 ml of MMAA 0.4 M NaCl medium were inoculated at 1/20 with overnight cultures prepared in LB-Difco medium. Cells were grown at 37 °C with shaking. After 330 min of culture, 3 ml samples were collected for each strain for the transcriptomic analysis. Cultures were stopped by the addition of two volumes of RNAprotect (Qiagen) and then centrifuged at $5000 \times g$ for 5 min. Pellets were isolated and immediately stored at −80 °C. Total RNA was extracted with the Direct-zol RNA MiniPrep kit (ZYMO RESEARCH). Ribosomal RNAs were specifically eliminated using the Ribo-Zero rRNA Removal kit (Illumina). The library and deep sequencing were performed by the GATC company on an Illumina HiSeq sequencer. Sequences were aligned with Bowtie2[98], indexed with Samtools[99], and counted with FeatureCount[100]. Normalization and statistical analysis were performed with R software[101] and Deseq script[102].

For the proteomic experiments, 100 ml samples were taken from triplicated cultures. Cells were washed in PBS and pelleted by centrifugation at $5000 \times g$ for 10 min at 4 °C. Pellets were resuspended in 150 μl of lysis buffer (PBS 1×, 1% protease inhibitor cocktail EDTA-free, Roche) and lysed by sonication (amplitude 70%, 3 pulses of 15 s separated by 15 s intervals). Two proteomics analyses were performed, named "proteomic $S_{15}$" and "proteomic $P_{150}$" respectively. First, for the proteomic $S_{15}$, lysates were centrifuged at $15,000 \times g$ for 30 min at 4 °C, and protein extracts were recovered and divided into 3 aliquots of 50 μl. The concentrations of the protein extracts were determined using a BCA assay (Pierce BCA Protein Assay, Thermo Fisher Scientific), and verification of the protein extracts was carried out on a 12.5% acrylamide SDS-PAGE gel stained with Coomassie brilliant blue R-250. For the proteomic analysis, 25 μg of each protein sample was diluted in a final volume of 25 μl for a final concentration of 8 M Urea/50 mM Ammonium Bicarbonate (AMBIC). Cysteines reduction was performed for 45 min at 37 °C in the presence of 4 mM of DL-Dithiothreitol (DTT) then alkylation step was carried out for 1 h at RT in the presence of 15 mM of iodoacetamide (IAA) then quenched with 2 mM of DTT for 10 min at RT, all the reactions were performed in the dark. The protein extracts were digested by Lys-C endoproteinase (500 ng for 3 h at 37 °C) and then by trypsin (500 ng at 37 °C overnight), both enzymes were purchased from Promega. The reactions were stopped by adding 1% trifluoroacetic acid (TFA) and peptides were desalted on Pierce C18 Spin Columns (Thermo Scientific) according to constructor specification. The desalted peptides were then dried under vacuum and resuspended in 100 μl of solvent A (0.1% (v/v) formic acid in 3% (v/v) acetonitrile) to a final concentration of 250 ng.μl$^{-1}$, before analysis by LC−MS/MS mass spectrometry. For the proteomic $P_{150}$, lysates were centrifuged at $150,000 \times g$ for 1 h at 4 °C. Pellets were resuspended in 200 μl of 1× PBS with 0.3% (v/v) SDS, and the concentrations of the protein extracts were determined using a BCA assay and verified on SDS-PAGE gel as for the $S_{15}$ samples. For the proteomic analysis, 25 μg of each sample were loaded on a 12.5% acrylamide SDS-PAGE gel and after a short (1 cm) migration, proteins were stained with Coomassie brilliant blue R-250. Lanes containing proteins were excised in two

bands and subjected to manual in-gel digestion with modified porcine trypsin (Trypsin Gold, Promega). Briefly, after destaining, bands were subjected to a 30 min reduction step at 56 °C using 10 mM DTT in 50 mM AMBIC prior to a 1-h cysteine alkylation step at RT both reactions were performed in the dark using 50 mM iodoacetamide in 50 mM AMBIC. After dehydration under vacuum, bands were re-swollen with 250 ng of trypsin in 200 μl 50 mM AMBIC and proteins were digested overnight at 37 °C. Supernatants were kept and peptides present in gel pieces were extracted with 1% (v/v) TFA. For each lane (meaning biological replicate), supernatants of the corresponding bands were pooled together and dried in a vacuum concentrator. Peptides extracts were then processed in the same way as protein extracts from proteomic $S_{15}$.

Mass spectrometry analyses were performed on a Q-Exactive Plus hybrid quadripole-orbitrap mass spectrometer (Thermo Fisher, San José, CA, USA) coupled to an Easy 1000 reverse phase nano-flow LC system (Proxeon) using the Easy nano-electrospray ion source (Thermo Fisher). Peptide mixtures were analyzed in triplicate. Briefly, 4 μl of peptide mixtures (1 μg) were loaded onto an Acclaim PepMap precolumn (75 μm × 2 cm, 3 μm, 100 Å; Thermo Scientific) equilibrated in solvent A and separated at a constant flow rate of 250 nl/min on a PepMap RSLC C18 Easy-Spray column (75 μm × 50 cm, 2 μm, 100 Å; Thermo Scientific) with a 90 min gradient (0–20% solvent B (0.1% (v/v) formic acid in acetonitrile) in 70 min and 20–37% solvent B in 20 min). Data acquisition was performed in positive and data-dependent Top10 modes. Full scan MS spectra (mass range m/z 400–1800) were acquired in profile mode with a resolution of 70,000 (at m/z 200) and MS/MS spectra were acquired in centroid mode at a resolution of 17,500 (at m/z 200).

*Data processing for label-free quantification* - Raw data were processed with the MaxQuant software package (http://www.maxquant.org, version 1.5.6.5)[103]. Protein identifications and target decoy searches were performed using the Andromeda search engine and the SwissProt database restricted to the *E. coli* K-12 taxonomy (release: 15/11/2018; 4477 entries) in combination with the Maxquant contaminants database (number of contaminants: 245). The mass tolerance in MS and MS/MS was set to 10 ppm and 20 mDa, respectively. Methionine oxidation and protein N-term acetylation were taken into consideration as variable modifications whereas cysteine carbamidomethylation was considered as fixed modification. Trypsin/P (including Pro cut) was selected as the cutting enzyme and 2 missed cleavages were allowed. Proteins were validated if at least 2 unique peptides having a protein FDR < 0.01 were identified. The setting "Match between runs" was also taken into consideration to increase the number of identified peptides. For quantification, we used unique and razor peptides with a minimum ratio count ≥2 unique peptides. Protein intensities were calculated by Delayed Normalization and Maximal Peptide Ratio Extraction[104] (MaxLFQ). Statistical analysis was done using Perseus software (https://maxquant.org/perseus, version 1.6.0.7) on LFQ intensities. Proteins belonging to contaminants and decoy databases were filtered. For each biological replicate, the median intensity of the two injected replicates was determined and proteins having quantitative data for all the biological replicates were considered for statistical analysis using a Benjamini−Hochberg test to control the False Discovery Rate (FDR). Principal component analysis of the $S_{15}$ proteomic result showed an outlier sample for one of the $\Delta ettA$ biological triplicates, therefore, it was excluded and the analysis was carried out on biological duplicates for the three strains.

### Protein expression and purification

EttA and EttA-EQ$_2$ proteins were expressed with an N-terminal hexahistidine-tag (His$_6$) from the pBAD plasmid under the control of an arabinose-inducible promoter in the strain MG1655. Protein production and purification were conducted following the previously published method[2] with the following changes. Cell lysis was performed by

3 successive passages in a cell disrupter (Constant Systems Ltd) at 2.5 kbar pressure. The initial purification step was realized on a Protino Ni-NTA column (Macherey-Nagel) then on a butyl-Sepharose FF (Cytiva) and finally monodispersed purified proteins were separated on a HiPrep 16/60 Sephacryl S-200 column (Cytiva). Purification buffers were the same as in Boël et al.[2]. The concentration of the protein was estimated using the BCA kit (Pierce BCA Protein Assay Kit - Thermo Fisher Scientific) according to the supplier's recommendations as well as by comparing the protein on a 12.5% acrylamide Coomassie-stained SDS-PAGE gel with BSA standards. The pure protein was frozen in liquid nitrogen and stored at −80 °C.

## Purification of *E. coli* MRE600 ribosomes

Highly purified ribosomes of the *E. coli* MRE600 strain, devoid of EttA protein (undetectable by Western blotting), were prepared by multiple centrifugations over sucrose cushions and gradients. Briefly, two liters of LB medium were inoculated with 20 ml of a saturated overnight culture of *E. coli* MRE600, grown at 37 °C under agitation to an $OD_{600}$ of 0.5. Cells were immediately harvested by centrifugation for 20 min, at 5000 g at 4 °C then resuspended and washed in buffer A (20 mM Tris pH 7.4, 10 mM Mg(OAc), 100 mM $NH_4$(OAc), 0.5 mM EDTA). All subsequent steps were performed at 4 °C. Cells were resuspended in buffer A supplemented with 0.1 mg.ml$^{-1}$ lysozyme, 6 mM β-mercaptoethanol, and 0.1% (v/v) protease inhibitor cocktail (Sigma–Aldrich), and lysed by 3 passages at 2.5 kBar using a cell disrupter (Constant Systems Ltd). Clarification of the lysate was then performed by two centrifugations of the extract at $22{,}000 \times g$ for 20 min at 4 °C. The supernatants were recovered and overlaid on an equal volume of 37.7% (w/v) sucrose cushion in buffer B (20 mM Tris pH 7.4, 10 mM Mg(OAc), 500 mM $NH_4$(OAc), 0.5 mM EDTA, 6 mM β-mercaptoethanol) and the samples were then centrifuged for 20 h, at $206{,}000 \times g$ at 4 °C in a Type 70 Ti rotor (Beckmann Coulter). Sucrose cushions were decanted and the clear ribosomal pellet was resuspended and incubated in buffer C (20 mM Tris pH 7.4, 7.5 mM Mg(OAc), 60 mM $NH_4$(OAc), 0.5 mM EDTA, 6 mM β-mercaptoethanol) under gentle agitation for 3 h, at 4 °C. Then 12 mg of crude ribosomes were loaded onto 10–40% (w/v) sucrose gradients prepared with buffer C and centrifuged for 16 h at $71{,}935 \times g$ at 4 °C in a SW28 rotor (Beckman Coulter). Gradients were fractionated on a Biocomp Piston Gradient Fractionator (Bio-Comp Instruments) and absorbance was measured at 254 nm. Fractions corresponding to the 70S ribosomes were pooled, washed, and concentrated in Amicon 50k (Merckmillipore) using Buffer C. The purified ribosomes were quantified by NanoDrop OneC UV-Vis Spectrophotometer (Thermo Scientific) checked on agarose gel, aliquoted, flash frozen in liquid nitrogen, and stored at −80 °C.

## In vitro transcription

The DNA templates used to generate mRNAs for in vitro translation assays and Toeprinting assays were PCR amplified with primers (Supplementary Table 7) from the pMMB plasmids containing the desired genes to be transcribed. The forward primer contains the T7 polymerase promoter sequence (5′-GCGAATTAATACGACTCACTATAGGG-3′). In vitro mRNA synthesis was then carried out at 37 °C for 4 h in the T7 RiboMAX Large Scale RNA Production System kit (Promega) according to the manufacturer's recommendations. At the end of the reaction, DNA templates were degraded by adding DNAse I for 15 min, and the mRNA transcripts were purified using TRIzol Reagent (Thermo Fisher Scientific) and Direct-zol RNA Miniprep kit (Zymo Research). The mRNAs were finally eluted in The RNA Storage Solution (Ambion, Thermo Fisher Scientific) and stored at −80 °C.

## In vitro translation assays

The PURExpress ΔRibosome In Vitro Protein Synthesis Kit (New England Biolabs) was used for in vitro translation assays. Reactions were performed in a final volume of 10 µl according to the instructions given by the manufacturer. This system consists of two solutions A and B in which all elements of the translational machinery are present except the ribosomes. Ribosomes purified from *E. coli* strain MRE600 were used at a final concentration of 2 µM per reaction. In a 10 µl translation reaction, the mRNA encoding each target gene fused with the *yfp* fluorescent reporter gene was used at a final concentration of 1 µM. The $His_6$-EttA protein previously diluted to 40 µM in the purification buffer and heated at 37 °C for 4 h (to increase the monomeric conformation) was added to the translation reactions at 5 µM final. As negative control the purified protein was replaced by the purification buffer. Reactions were transferred to a 384-well plate (Sigma–Aldrich), incubated for 150 min in the CLARIOstar plate reader (BMG Labtech) and translation of the mRNA was monitored every 2 min by measuring YFP fluorescence as previously described. All the translation assays were performed in triplicate for each condition.

## Toeprinting assays

Assays were performed using the PURExpress ΔRibosome (New England Biolabs) according to manufacturer instructions, to determine stalled-ribosome sequence motifs on the target mRNAs. Purified $His_6$-EttA or $His_6$-EttA-$EQ_2$ proteins were added to in vitro translation reactions at final concentrations of 2 µM and 5 µM respectively to evaluate the effect of the ABC-F on the stalled ribosomes. In the reactions in which the effect of antibiotics was tested, the antibiotics (thiostrepton or retapamulin) were added first so as to have a final concentration of 50 µM in the tubes and then dried using a SpeedVac vacuum concentrator (Thermo Fisher Scientific). Purified ribosomes from MRE600 were added to the reactions at a final concentration of 2 µM and mRNA templates at 1 µM. Reactions were then incubated at 37 °C for 15 min and 2 pmoles of CY5-labeled primer (Supplementary Table 7) complementary to NV1 sequence[105] (5′ GGTTATAATGAATTTTGCTTATT 3′) were added and incubated immediately for 5 min at 37 °C. Finally, reverse transcription reactions were performed by adding 0.5 µl (corresponding to 5 U) of AMV Reverse Transcriptase (Promega), 0.1 µl dNTP mix (10 mM), 0.4 µl Pure System Buffer (5 mM K-phosphate pH 7.3, 9 mM Mg(OAc), 95 mM K-glutamate, 5 mM $NH_4$Cl, 0.5 mM $CaCl_2$, 1 mM spermidine, 8 mM putrescine, 1 mM DTT) for each reaction and incubated at 37 °C for 20 min. Once Cy5-labeled cDNA was generated, the mRNAs were degraded by the addition of 0.5 µl of 10 M NaOH and incubation at 37 °C for 20 min. All the reactions were neutralized by the addition of 0.7 µl of 7.5 M HCl followed immediately by the addition of 20 µl of toeprint resuspension buffer (300 mM Na-acetate pH 5.5, 5 mM EDTA, and 0.5% SDS (v/v)). Using QIAquick Nucleotide Removal kit (Qiagen), cDNAs were purified according to supplier instructions and then dried and resuspended in 6 µl of toeprint loading dye (95% formamide, 250 µM EDTA, and 0.25% (w/v) bromophenol blue). Sanger sequencing reactions were also performed on the DNA templates that were used for mRNA synthesis for the toeprint assays. Polymerization reactions were prepared in 20 µl with ddNTPs (625 µM ddCTP/ddTTP/ddATP or 50 µM ddGTP), 0.025 U Taq Pol (New England Biolabs), 1× Thermo Pol Buffer (New England Biolabs), 5 nM DNA matrix, 75 nM CY5-labeled primer 137 and 40 µM of each dNTP. After amplification by PCR, 20 µl formamide dye was added to each sequencing reaction and all of the samples were denatured at 80 °C for 3 min before being loaded onto the acrylamide/bis-acrylamide/urea sequencing gel. Cy5-labeled cDNAs were detected by fluorescent mode using a Typhoon FLA 9500 (GE Healthcare Life Sciences) Gel Scanner and LPR Ch.2 filter and 635 nm laser.

## Analytical gel-filtration and static light-scattering analyses

Protein samples were injected onto a Yarra 3 µm SEC-2000 (Phenomenex) running at RT in 150 mM NaCl, 5% (v/v) glycerol, 20 mM Tris-HCl, pH 7.2. The column effluent was monitored with multi-angle light-scattering detector (miniDAWN treos) and refractive index (Optilab T-rEX) detectors from Wyatt Technologies.

## Image analyses and preparation

Western and northern-blot signals were quantified using Fiji[88] software. Figures of the structure of EttA-EQ$_2$ in complex with 70S IC complex (EMD-29398)[18] and structure (6ENJ)[68] of the polyproline-stalled ribosome in complex with EF-P were prepared using PyMOL Molecular Graphics System (Schrödinger) and the structures were aligned on the domain V of the 23S rRNA.

## Statistics and reproducibility

All the experiments presented have been reproduced at least twice with the same results. The details of the different statistical analyses presented herein are the following:

*Bacteria growth rate* - We extracted the early exponential phase of the growth curves, and then the doubling time was calculated by fitting the curves to the Malthusian growth equation;

$$Y = Y_o e^{(kt)} \tag{1}$$

$$\text{doubling time} = \frac{\ln(2)}{t} \tag{2}$$

Where $Y$ is the population, $Y_O$ the starting population, $k$ the rate, and $t$ the time. The lag was determined by measuring the delta between the tangent interception with the time axes of the WT strain and compared strain.

*Determination of the fluorescence ratio* - Calculation of fluorescence ratios were done by first calculating the ratio of the fluorescence to OD$_{600}$ (Fluo/OD) of each triplicate to obtain the normalized fluorescence (FluoN), then the average for the WT, $\Delta ettA$ and $C_{ettA}$ strains was corrected by subtracting the average of the FluoN for the WT pMMB-Ø strain (WT strain carrying an empty pMMB plasmid) (FluoN$_{control}$) to obtain the corrected normalized fluorescence (FluoNC). Finally, we calculated the ratios of the FluoNC for the $\Delta ettA$ and the $C_{ettA}$ strains compared to the WT strain. The propagation of uncertainty was determined by the function:

$$f = A - B \tag{3}$$

Where A is the FluoN of the strain WT, $\Delta ettA$ or $C_{ettA}$ and B the FluoN$_{control}$. The resulting standard deviation of the FluoNC was calculated using the equation:

$$\sigma_f = \sqrt{\sigma_A^2 + \sigma_B^2 - 2\sigma_{AB}} \tag{4}$$

Where $\sigma_A$ and $\sigma_B$ are the standard deviations of the variables $A$ and $B$ and $\sigma_{AB}$ the covariance.

Finally, the standard deviation of the ratio ($\sigma_r$) is:

$$\sigma_r = \frac{FluoNC_{WT}}{\sigma_f} \tag{5}$$

For the fluorescence reporters for *mgtL* or *mgtA* the normalization was done on the WT strain in the 2 mM Mg$^{2+}$ culture condition.

*Protein sequence motif analyses* - Presence and position of the proteins containing the motif E/D X{0,1} E/D within the first 50 aa of the N-terminal part of the sequence were determined in the *E. coli* proteome using the Ecocyc database[106] and mapped to the proteomic quantification ($S_{15}$) data (with the EttA targets presented in this study previously excluded). Then the log$_2$ ratios of the protein quantification ratios for $\Delta ettA$ vs WT and $C_{ettA}$ vs WT were binned into six bins: absence of the motif ($N = 488$ for $\Delta ettA$ vs WT and $N = 476$ for $C_{ettA}$ vs WT), motif between residues 1 and 10 ($N = 121$ for $\Delta ettA$ vs WT and $N = 116$ for $C_{ettA}$ vs WT), motif between residues 11 and 20 ($N = 204$ for $\Delta ettA$ vs WT and $N = 200$ for $C_{ettA}$ vs WT), motif between residues 21 and 30 ($N = 184$ for $\Delta ettA$ vs WT and $N = 176$ for $C_{ettA}$ vs WT), motif between residues 31 and 40 ($N = 184$ for $\Delta ettA$ vs WT and $N = 177$ for $C_{ettA}$ vs WT) and motif between residues 41 and 50 ($N = 152$ for $\Delta ettA$ vs WT and $N = 150$ for $C_{ettA}$ vs WT). Violin plots were generated using Prism 8 (GraphPad) and *p*-values were calculated using a two-tailed Mann–Whitney test.

## Reporting summary

Further information on research design is available in the Nature Portfolio Reporting Summary linked to this article.

## Data availability

RNA-sequencing data are available through the EBI ArrayExpress database (https://www.ebi.ac.uk/biostudies/arrayexpress/studies) under accession code: E-MTAB-13458. Proteomic data are available through the massIVE Repository (https://massive.ucsd.edu/ProteoSAFe/static/massive.jsp) under accession code: MSV000093143. Source data are provided with this paper.

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

## Acknowledgements

F.O., T.O., and G.B. were supported by funds from the CNRS (UMR8261), Université Paris Cité, the LABEX program (DYNAMO ANR-11-LABX-0011), and two ANR grants (EZOtrad/ANR-14-ACHN-0027 and ABC-F_AB/ANR-18-CE35-0010). F.O. received a fellowship from the Edmond de Rothschild Foundation. This work was also supported by the EQUIPEX CAC-SICE (ANR-11-EQPX-0008), through funding of the Proteomic Platform of IBPC. We express our great gratitude to Pr. Hirotada Mori for his gifts of the Keio collection and the YFP fusion strains collection. We thank Alexandre Pozza for his help with SEC-MALS experiments. The authors thank Jackie Plumbridge, Nicolas Biais, and John F. Hunt for their advice and help.

## Author contributions

F.O. constructed all the strains, performed most of the experiments, T.O. and S.N. performed some bacterial cultures and sample preparations, S.N. performed the northern-blot experiments, M.H. performed the proteomic analysis, A.O.N. generated early results used for this study. F.O., G.B., and L.M. analyzed the data and prepared the figures, G.B. designed the research program. G.B., L.M., and F.O. wrote the manuscript in consultation with the other authors.

## Competing interests

The authors declare no competing interests.
