## [Peer Review File · Nature Communications]

Global regulation via modulation of ribosome pausing by the ABC-F protein EttAREVIEWER COMMENTS

Reviewer #1 (Remarks to the Author):

This manuscript by Ousalem et al. provides insights into the function of EttA, a member of the poorly understood ABC-F family of proteins, in protein synthesis. The authors provide a series of careful studies of the phenotypes of *E. coli* cells lacking EttA. They identify a salt sensitivity under certain nutrient conditions and are able to tie that phenotype to differences in translation of a few genes in the TCA cycle and glyoxylate shunt with or without EttA. These results show that EttA suppresses ribosome drop-off on poly-acidic sequences early in the protein sequence. The authors can tie this mechanism to the phenotype of the knockout. Other work on ABC-F proteins in bacteria is starting to point in a similar direction (in recent pre-prints). What sets this manuscript apart is the thorough characterization of how loss of EttA affects the biology of the cell. I have a few suggestions about presentation and firming up some of the biochemical data, but overall I find the results very compelling.

Specific comments:

Fig 1b: Regarding the transcriptional changes in EttA, it looks like EttA might be driven by an RpoS promoter—this might be part of the regulation of these genes under stress conditions through the action of EttA.

Fig 1c: The YFP fusion data reports on how well the fusion proteins are translated with or without EttA. Perhaps this would best be put into a different figure, since the other data in this figure have to do with the TCA cycle and how well the three strains grow under various conditions (e.g. phenotypes and not the translation of reporters). Right now there is just too much going on all at once, and the YFP fusions don't seem to belong here. Perhaps they could be added to Fig 2.

Fig 2c/d: In the toeprinting data in these figures, different results are seen with AceB and YjbJ. There is a pause in YjbJ but not AceB without any added factors. Then when the EQ2 mutant is added, it creates a pause in AceB but not YjbJ. These results appear contradictory. The authors suggest that in the AceB case, without factors, a pause results in drop-off, which we cannot see in a toeprint assay. One of the shortcomings of the paper is that they don't really prove that drop-off is happening (although based on the earlier work by Chadani, I believe that it is). I can think of two ways to look for drop-off here: 1) Repeat the toeprinting assays, but trap the ribosomes at a site downstream of the DE pause site, either at a stop codon (with no release factors in the reaction) or at an Ile codon with mupirocin added to the reaction (see papers by Shura Mankin for the use of this "trap" codon in toeprinting). This would allow the authors to see all the ribosomes loaded on the mRNA, and if there is drop-off, then there would be fewer ribosomes at the "trap" codon. 2) Alternatively they could look for peptidyl-tRNA that has fallen off of the ribosome, and is therefore sensitive to RNases, as Chadani has done in his drop-off papers.

Fig 3a: The red and green arrows are unnecessary and make it harder to read.

Fig 3: I can follow the logic in the left part of Fig 3a about why lower expression of *aceB* in the cells lacking EttA leads to more flux through the *gcl* pathway and better survival in media with glyoxylate. The knockouts seem to back up their inferences. But I have a very hard time following the logic on page 11 (results) and the data shown in the right part of Fig 3a having to do with the 0.4 M NaCl stress. Please clarify. I don't see what these genes have to do with the stress of high salt.

Fig 4c: Is it possible the labels are switched? Why is this result in vitro the opposite of what we see in Fig 4b in vivo?

Several of the problematic amino acid sequences have Pro residues as well as acidic residues. Does addition of EFP to the in vitro translation reactions restore some level of translation through these sequences (like EttA does)?

Reviewer #2 (Remarks to the Author):

The paper of Ousalem et al. explores the role of the ABC-F protein EttA in mRNA translation and

gene regulation in *E. coli*. By using proteomics, authors show that the lack of EttA leads to changes in cellular levels of several proteins, including the ones involved in central metabolism. Subsequent biochemical and in vivo studies showed that EttA modulates translation of several functional and regulatory genes that encode negatively charged amino acids within the 5' -proximal codons. The authors propose the model according to which EttA can either stabilize the ribosome prone to dissociation at the poly-acidic sequences or facilitate the resolution of the stalled translation complex.

Critique

The functional role of ABC-F proteins is a hot topic. The work of Ousalem et al. provides a notable advance to our understanding of the operation of one of such proteins and thus contributes significantly to this burgeoning field of research. The work is solid and is based on a healthy combination of complementary in vivo and in vitro approaches. The experiments are properly controlled, and the results are convincing. However, the paper is not an easy read. Many arguments are convoluted and some figures (in particularly those in the extended data) are overloaded. Streamlining the text and simplifying the figures would make the work much more impactful. I am also not convinced that dedicating a significant part of the paper to discussing the effect of EttA upon the central metabolism is beneficial. In my opinion, shortening the corresponding sections, while focusing the story more on the effects of EttA on translation, would be beneficial.

The rest of my comments are more of a cosmetic nature:

ED Fig. 1. What does 'MW' over the first lane of agarose gels in panel a stand for? If 'molecular weight', then it is incorrect.

ED Fig. 1. The bottom part of panel 'a' is terribly confusing. Could it be simplified?

p.3 (middle) and Fig. 1. The addition of the shaded panels in Fig. 1a additionally complicates an already complicated panel.

p.6. The discussion of differences in protein abundance and its correlation with mRNA is rather complicated. Clearly separating these two aspects is highly advisable. Furthermore, if the authors really want to discuss the effect of EttA on translation, they should operate with the Translation Efficiency (TE) parameter which can be deduced from the comparison of Ribo-seq and RNA-seq data.

p.9 top and Fig. 2C. According to my counting of the bands in the sequencing ladder, the toeprint bands at the start codon and at the "arrested elongation complex" site are separated not by 24 nt, but by 26 or 27 nt. Therefore, the ribosome seems to be arrested not at the GAU (Asp) codon, but at the next codon: GAA (Glu) codon. Similarly, in the gel in Fig. 2d, not seven, but 4 (maximum 5 nt) separate the start codon toeprint band and the arrested complex toeprint band; this puts the arrested ribosome squarely at the 3d (Lys) codon, not at the 4th (Asp) codon.

Reviewer #3 (Remarks to the Author):

I have been invited by Nature Communications to review the proteomics studies performed in the manuscript entitled 'Global regulation via modulation of ribosome pausing by EttA'. In this manuscript authors have performed the characterization of the two fractions S15 and P150. The statistical analysis and the figures included as extended data are correct, the methodology part describing these studies is very detailed and just minor edits are suggested below in the point-by-point changes and questions list included in this revision; nonetheless, in the results section the proteomics results are merely mentioned in a generalistic sentence. It is not clear reading the results section why authors performed the proteomics studies, what type of results were obtained from these studies and what were the main findings. My suggestion is that authors should expand that section, giving proper credit to these studies.

Point-by-point changes and questions:

Methods section – RNA-seq and proteomics experiments:

- Close the parenthesis when defining the components of the lysis buffer.
- Revise that all the units of centrifugation x g are correctly placed.
- Specify the time and temperature of the reduction and alkylation reactions.
- Use the '.' in this expression: 250 ng.µl- 1 or write ng/µl.
- Add LC-MS/MS into the sentence:before analysis by MS / MS mass spectrometry.
- Why authors did a complete run with 12.5% acrylamide SDS-PAGE gel for proteomics S15 experiment and a short migration of 1cm for the P150?
- Why authors used Lys-C endoproteinase and trypsin Gold from Promega in the proteomics S15 experiment and just porcine trypsin for the P150?
- Change that sentence: 'Briefly, peptide mixtures were analyzed in triplicate. 4 µl of peptide mixtures were....' By 'Peptide mixtures were analyzed in triplicate. Briefly, 4 µl of peptide mixtures were....'.
- When the strategy used for database search is explained Endoproteinase Lys-C is not considered, and this enzyme cleaves after Proline and trypsin alone does not, thus it should be considered if used during digestion.
- I feel that although methodology is very detailed for proteomics studies, in results the proteomics studies are briefly mentioned, and much more importance is given in this section to genes and not to proteins. Thus, I suggest to expand this explanation in results section. What are the differences in protein expression observed? Authors mention that in the proteomic studies differences were observed but more details are needed.

Paris, Friday, May 3, 2024

To the reviewers :

General response to the reviewers:

We would first like to express our gratitude to the reviewers for their careful review of our manuscript. We appreciate the insightful points raised on how to improve our work. Following the reviewers' advice, we have made substantial revisions to our manuscript to enhance clarity and address their comments. We have uploaded a revised version of the manuscript with changes highlighted using Microsoft Word's track changes option.

The major changes are summarized below:

- The original Extended Data and Supplementary Information documents have been consolidated into one supplementary information file to adhere to the Nature Communications format.
- The Extended Data Figures are now Supplementary Figures.
- The new Supplementary Figure 2 corresponding to the original Extended Data Figure 2 has been simplified. Some content (panels d, f, g and h) has been transferred in a new figure in the main manuscript (Fig. 2) and the panel, e, showing the ratio of protein quantification for the ΔettA and C_{ettA} strains compared to the WT, has been removed as it contained redundant information.
- The main original Figures 2, 3, 4 and 5 are now Fig 3, 4, 5 and 6 respectively.
- All the figures have been modified to improve clarity and implement reviewers' comments.

Point-by-point responses to reviews of NCOMMS-24-05419-T by Ousalem *et al.* (reviewers 1- 3):

• **Reviewer #1 (Remarks to the Author):**

This manuscript by Ousalem *et al.* provides insights into the function of EttA, a member of the poorly understood ABC-F family of proteins, in protein synthesis. The authors provide a series of careful studies of the phenotypes of *E. coli* cells lacking EttA. They identify a salt sensitivity under certain nutrient conditions and are able to tie that phenotype to differences in translation of a few genes in the TCA cycle and glyoxylate shunt with or without EttA. These results show that EttA suppresses ribosome drop-off on poly-acidic sequences early in the protein sequence. The authors can tie this mechanism to the phenotype of the knockout. Other work on ABC-F proteins in bacteria is starting to point in a similar direction (in recent pre-prints). What sets this manuscript apart is the thorough characterization of how loss of EttA affects the biology of the cell. I have a few suggestions about presentation and firming up some of the biochemical data, but overall I find the results very compelling.

Authors' main response to Reviewer #1: We thank the reviewer for his/her insightful and critical judgment on our work. It is gratifying to know that the reviewer shares our appreciation for the physiological aspect of the manuscript and acknowledges the significance of linking EttA's mechanism of action to the phenotype. We believe that this approach is instrumental in advancing our understanding of EttA's function within the cell.

Specific comments:

Fig 1b: Regarding the transcriptional changes in EttA, it looks like EttA might be driven by an RpoS promoter—this might be part of the regulation of these genes under stress conditions through the action of EttA.

Authors' response: We concur with the reviewer and had previously suggested, in the last paragraph of the discussion, the possibility of EttA transcription being driven by an RpoS promoter. Prompted by the reviewer's comment, we have now included this point also in the results section and further discussed how it may contribute to regulation during stress in the discussion section of the revised manuscript.

Fig 1c: The YFP fusion data reports on how well the fusion proteins are translated with or without EttA. Perhaps this would best be put into a different figure, since the other data in this figure have to do with the TCA cycle and how well the three strains grow under various conditions (e.g. phenotypes and not the translation of reporters). Right now there is just too much going on all at once, and the YFP fusions don't seem to belong here. Perhaps they could be added to Fig 2.

Authors' response: We thank the reviewer for this suggestion, we have now moved the YFP fusion originally presented in Fig. 1 to a new Fig 2 which also contains the proteomic results. Therefore Fig. 1 is now only about the growth phenotypes and the new Fig.2 about identification and validation of EttA's targets.

Fig 2c/d: In the toeprinting data in these figures, different results are seen with AceB and YjbJ. There is a pause in YjbJ but not AceB without any added factors. Then when the EQ2 mutant is added, it creates a pause in AceB but not YjbJ. These results appear contradictory. The authors suggest that in the AceB case, without factors, a pause results in drop-off, which we cannot see in a toeprint assay. One of the shortcomings of the paper is that they don't really prove that drop-off is happening (although based on the earlier work by Chadani, I believe that it is). I can think of two ways to look for drop-off here: 1) Repeat the toeprinting assays, but trap the ribosomes at a site downstream of the DE pause site, either at a stop codon (with no release factors in the reaction) or at an Ile codon with mupirocin added to the reaction (see papers by Shura Mankin for the use of this "trap" codon in toeprinting). This would allow the authors to see all the ribosomes loaded on the mRNA, and if there is drop-off, . 2) Alternatively they could look for peptidyl-tRNA that has fallen off of the ribosome, and is therefore sensitive to RNases, as Chadani has done in his drop-off papers.

Authors' response: We acknowledge that the toeprint observations with AceB and YjbJ yield different results; however, we do not view these results as contradictory. Instead, they suggest that following ribosome pausing, two different outcomes may occur: in the case of AceB, ribosome dissociation, and in the case of YjbJ, ribosome stalling.

We have already endowed the idea of direct demonstration of the drop-off but, one significant challenge in these assays is the small difference observed *in vitro* between conditions with or without EttA, which limits the sensitivity of the analysis. Detecting small changes is challenging, particularly since detecting stalling *via* toe-printing was already difficult. Consequently, we were concerned that utilizing the trap codon approach would be challenging, given the limited sensitivity of quantifying a small reduction in signal variation for the trapping codon.

Therefore, we shifted our focus to drop-off assays (**Figure 1** below). We were able to observe a signal corresponding to peptidyl-tRNA (**Figure 1b**; below), visible only in constructs containing the first 10 amino acids of AceB fused to YFP and not in equivalent constructs where the two acidic residues are mutated (AceB-NQ-10aa:YFP). The addition of EttA increased the amount of full protein translated and

reduced the ratio of full translation versus peptidyl-tRNA drop-off (**Figure 1c** below). We have repeated these results twice, but the signal in all cases is relatively weak.

Fig 3a: The red and green arrows are unnecessary and make it harder to read.

Authors' response: This figure, now labeled as Fig. 4, has been reorganized in an attempt to simplify it. The red and green arrows have been removed.

Fig 3: I can follow the logic in the left part of Fig 3a about why lower expression of *aceB* in the cells lacking *EttA* leads to more flux through the *gcl* pathway and better survival in media with glyoxylate. The knockouts seem to back up their inferences. But I have a very hard time following the logic on page 11 (results) and the data shown in the right part of Fig 3a having to do with the 0.4 M NaCl stress. Please clarify. I don't see what these genes have to do with the stress of high salt.

Authors' response: We acknowledge that this section was complicated and not well explained. We have rewritten this section to make it easier to understand. Our deletion analysis has demonstrated that *aceA* is necessary for growth in the presence of salt. Therefore, we believe that the lower expression of *aceA*, in the Δ *ettA*, strain is partly responsible for the phenotype. In the Δ *ettA* strain, the reduced expression of *aceA* should decrease the metabolic flux toward the glyoxylate shunt and tartronate semialdehyde pathways, thus promoting it towards the ICD enzyme and the carbon dioxide-producing steps of the TCA cycle. Accordingly, a deletion that impairs the tartronate semialdehyde pathway (Δ *gcl*) has a strong deleterious effect on the WT and complemented strains, but not on the Δ *ettA* strain. Conversely, deletions of genes involved in the carbon dioxide-producing steps of the TCA cycle are more deleterious for the Δ *ettA* strain.

Fig 4c: Is it possible the labels are switched? Why is this result in vitro the opposite of what we see in Fig 4b in vivo?

Authors' response: Absolutely, we express our gratitude to the reviewer for their meticulous review and for bringing this mistake to our attention. We have rectified it in the revised version of the manuscript.

Several of the problematic amino acid sequences have Pro residues as well as acidic residues. Does addition of EFP to the in vitro translation reactions restore some level of translation through these sequences (like EttA does)?

Authors' response: We didn't explore this avenue EF-P because most of the target contain only one proline or none. The EttA's target with the most of proline is MgtL and Y. Chadani et al., have shown (Chadani Y, *et al.* 2017. Mol Cell 68: 528-39 e5. <https://doi.org/10.1016/j.molcel.2017.10.020>, Figure 4C) that EF-P does not alleviate peptidyl-tRNA drop-off or IRD of MgtL.

- **Reviewer #2 (Remarks to the Author):**

The paper of Ousalem et al. explores the role of the ABC-F protein EttA in mRNA translation and gene regulation in *E. coli*. By using proteomics, authors show that the lack of EttA leads to changes in cellular levels of several proteins, including the ones involved in central metabolism. Subsequent biochemical and in vivo studies showed that EttA modulates translation of several functional and regulatory genes that encode negatively charged amino acids within the 5' -proximal codons. The authors propose the model according to which EttA can either stabilize the ribosome prone to dissociation at the poly-acidic sequences or facilitate the resolution of the stalled translation complex.

Critique

The functional role of ABC-F proteins is a hot topic. The work of Ousalem et al. provides a notable advance to our understanding of the operation of one of such proteins and thus contributes significantly to this burgeoning field of research. The work is solid and is based on a healthy combination of complementary in vivo and in vitro approaches. The experiments are properly controlled, and the results are convincing.

However, the paper is not an easy read. Many arguments are convoluted and some figures (in particularly those in the extended data) are overloaded. Streamlining the text and simplifying the figures would make the work much more impactful. I am also not convinced that dedicating a significant part of the paper to discussing the effect of EttA upon the central metabolism is beneficial. In my opinion, shortening the corresponding sections, while focusing the story more on the effects of EttA on translation, would be beneficial.

Authors' main response to Reviewer #2: We thank the reviewer for this input and judgment on our work, and we appreciate that the reviewer found our manuscript to represent a notable advance in the understanding of ABC-F function. We also value the recognition of our efforts to combine *in vivo* and *in vitro* approaches.

We have significantly revised the text of our manuscript with the hope of facilitating reading and comprehension. As suggested by the reviewer, we have simplified the Extended Data figures, with particular attention to simplifying ED Fig. 1, which is now referred to as SI Fig. 1.

Regarding the discussion about the physiological implications of EttA on central metabolism, we consider it essential because it demonstrates that even small changes in genes expression modulated by EttA have a real physiological impact. This suggests that EttA has an important regulatory function within the cell.

Furthermore, our work has demonstrated the importance of the glyoxylate shunt and tartronate semialdehyde pathways for stress adaptation. We understand that including these results in the manuscript solidifies the findings, but it also considerably complicates the manuscript. We have considered the idea of presenting the physiological aspect of our study in another manuscript, but we have decided that it would be impractical and would dilute the overall impact of our study.

The rest of my comments are more of a cosmetic nature:

ED Fig. 1. What does 'MW' over the first lane of agarose gels in panel a stand for? If 'molecular weight', then it is incorrect.

Authors' response: Thanks for noticing this point; we have corrected it accordingly

ED Fig. 1. The bottom part of panel 'a' is terribly confusing. Could it be simplified?

Authors' response: We have simplified ED Fig. 1 (now SI Fig. 1) by removing some content from this figure and transferring it to a new figure (Fig. 2). Additionally, we have added some illustrations to the bottom of panel a in an attempt to enhance the comprehension of the reported experiments.

p.3 (middle) and Fig. 1. The addition of the shaded panels in Fig. 1a additionally complicates an already complicated panel.

Authors' response: We have reorganized the shaded panels in Fig. 1a to make the figure clearer. However, we did not remove them because we believe they help the reader understand the shaded panels (heatmaps) representation that we later used in panel c of this figure and in Fig. 4.

p.6. The discussion of differences in protein abundance and its correlation with mRNA is rather complicated. Clearly separating these two aspects is highly advisable. Furthermore, if the authors really want to discuss the effect of EttA on translation, they should operate with the Translation Efficiency (TE) parameter which can be deduced from the comparison of Ribo-seq and RNA-seq data.

Authors' response: We did not have Ribo-seq data in our study; instead, we present proteomic and RNA-seq data. Therefore, we prefer not to report a Translation Efficiency value because we lack the necessary data for that analysis. However, we have reformulated the text to simplify the explanations and added a new figure in the main body (Fig. 2). This figure contains a bar plot presenting the transcriptomic and proteomic values of the selected targets, which were previously included in the Extended Data figures.

p.9 top and Fig. 2C. According to my counting of the bands in the sequencing ladder, the toeprint bands at the start codon and at the "arrested elongation complex" site are separated not by 24 nt, but by 26 or 27 nt. Therefore, the ribosome seems to be arrested not at the GAU (Asp) codon, but at the next codon: GAA (Glu) codon. Similarly, in the gel in Fig. 2d, not seven, but 4 (maximum 5 nt) separate the start codon toeprint band and the arrested complex toeprint band; this puts the arrested ribosome squarely at the 3d (Lys) codon, not at the 4th (Asp) codon.

Authors' response: We express our gratitude to the reviewer for highlighting this problematic point and bringing to our attention a mistake we made. In Fig. 2c (now Fig. 3c), there is a distance of 25 nucleotides between the toeprint of the arrested elongation complex and the initiation complex

(taking into consideration the lower band observed with Retapamulin or with EttA-EQ2). If we consider 1 for the A of AUG that brings it to G of the GAU codon. Another interpretation of the toeprint involves averaging 15-16 nucleotides between the ribosomal P site and the toeprinting signal, placing the GAU codon at the P site.

In Fig. 2d (now Fig. 3d), we made a mistake; indeed, there are only 4 nucleotides separating the initiation toeprint and the stalled one. Using the same method to estimate the P-site codon as described above (15-16 nucleotides from the toeprinting signal), this would place, as proposed by the reviewer, the Lysine as the P-site codon.

We have appropriately corrected the text and added the sequence below the toeprint in Fig. 2 to clarify the location of the toeprinting signal and our interpretation of the P-site codon.

- **Reviewer #3 (Remarks to the Author):**

I have been invited by Nature Communications to review the proteomics studies performed in the manuscript entitled 'Global regulation via modulation of ribosome pausing by EttA'. In this manuscript authors have performed the characterization of the two fractions S15 and P150. The statistical analysis and the figures included as extended data are correct, the methodology part describing these studies is very detailed and just minor edits are suggested below in the point-by-point changes and questions list included in this revision; nonetheless, in the results section the proteomics results are merely mentioned in a generalistic sentence. It is not clear reading the results section why authors performed the proteomics studies, what type of results were obtained from these studies and what were the main findings. My suggestion is that authors should expand that section, giving proper credit to these studies.

Authors' main response to Reviewer #3: We thank the reviewer for reviewing the proteomics studies presented in the manuscript. We appreciate the great effort made to carefully review our method sections and for catching some error for which we apologize. We had originally submitted this manuscript to a journal with a very restrictive format. For this reason, we weren't able to dedicate more space to the proteomics studies that did allow us to identify the targets of EttA. We have now reorganized the manuscript to bring the proteomics analysis in the main part of the manuscript. A new figure in main, Fig.2 presents all the data of the proteomic that were previously in ED Fig. 2.

Point-by-point changes and questions:

Methods section – RNA-seq and proteomics experiments:

- Close the parenthesis when defining the components of the lysis buffer.

Authors' response: Done.

- Revise that all the units of centrifugation x g are correctly placed.

Authors' response: Thanks for spotting this problem which is now fixed in the revised manuscript.

- Specify the time and temperature of the reduction and alkylation reactions.

Authors' response: We have added this information to the methods section.

- Use the '·' in this expression: 250 ng·µl- 1 or write ng/µl.

Authors' response: Done.

- Add LC-MS/MS into the sentence:before analysis by MS / MS mass spectrometry.

Authors' response: Done.

- Why authors did a complete run with 12.5% acrylamide SDS-PAGE gel for proteomics S15 experiment and a short migration of 1cm for the P150?

Authors' response: The complete run on the SDS-PAGE gel was conducted to verify the integrity of the samples. The short run for the P150 samples was performed to prepare the samples for in gel trypsinolysis. Indeed, the brief electrophoresis helps in the preparation of the protein extract by removing the detergent during the washes. However, P150 samples were also run completely on an 12.5% acrylamide SDS-PAGE gel. We have corrected this point and clarified the two different runs used in the methods part of the manuscript.

- Why authors used Lys-C endoproteinase and trypsin Gold from Promega in the proteomics S15 experiment and just porcine trypsin for the P150?

Authors' response: The protocol for digestion in solution, used for S15 samples, includes an additional Lys-C endoproteinase to improve the cleavage after Lysine. For the in-gel digestion used for P150 samples, this step is not necessary.

- Change that sentence: 'Briefly, peptide mixtures were analyzed in triplicate. 4 µl of peptide mixtures were....' By 'Peptide mixtures were analyzed in triplicate. Briefly, 4 µl of peptide mixtures were....'.

Authors' response: Done

- When the strategy used for database search is explained Endoproteinase Lys-C is not considered, and this enzyme cleaves after Proline and trypsin alone does not, thus it should be considered if used during digestion.

Authors' response: We thank the reviewer for specifying that point. We used the MaxQuant digestion enzyme by default, which is Trypsin/P. This setup considers cleavage after proline, as our trypsin enzyme can also cleave after proline. We have corrected that point in the methods section of the manuscript.

- I feel that although methodology is very detailed for proteomics studies, in results the proteomics studies are briefly mentioned, and much more importance is given in this section to genes and not to proteins. Thus, I suggest to expand this explanation in results section. What are the differences in protein expression observed? Authors mention that in the proteomic studies differences were observed but more details are needed.

Authors' response: We concurred with the reviewer's feedback. As explained in our main response to the reviewer, the format restrictions of the original journal to which we submitted the manuscript necessitated the condensation of our results. Now, with more space available, we have revised the manuscript to incorporate additional information about the proteomic results into the main body of the manuscript.

We would like to reiterate our appreciation for the insightful work done by the reviewers. The comments have led to substantial improvements in both the rigor and the clarity of our manuscript.

We remain at your disposal for any questions or if any further information or improvement is required.

Best regards,

Grégory Boël, PhD
Principal Investigator
CNRS / Université de Paris, UMR8261
Institut de Biologie Physico-Chimique
13 rue Pierre et Marie Curie, 75005, Paris, France

REVIEWERS' COMMENTS

Reviewer #1 (Remarks to the Author):

The authors have addressed my concerns raised in the first round of review. The rearrangement of the data in Fig 1 and 2 makes the paper clearer and the new discussion of salt tolerance on page 11 is easier to follow.

Reviewer #2 (Remarks to the Author):

The authors have successfully addressed my questions and concerns.

Reviewer #3 (Remarks to the Author):

The authors have successfully addressed all my concerns regarding the proteomics experiments.